# Differential Privacy Dynamics of Langevin Diffusion and Noisy Gradient Descent

**Rishav Chourasia**[*], **Jiayuan Ye**[*], **Reza Shokri**
Department of Computer Science, National University of Singapore
{rishav1, jiayuan, reza}@comp.nus.edu.sg

## Abstract

What is the information leakage of an iterative randomized learning algorithm about its training data, when the internal state of the algorithm is *private*? How much is the contribution of each specific training epoch to the information leakage through the released model? We study this problem for noisy gradient descent algorithms, and model the *dynamics* of Rényi differential privacy loss throughout the training process. Our analysis traces a provably *tight* bound on the Rényi divergence between the pair of probability distributions over parameters of models trained on neighboring datasets. We prove that the privacy loss converges exponentially fast, for smooth and strongly convex loss functions, which is a significant improvement over composition theorems (which over-estimate the privacy loss by upper-bounding its total value over all intermediate gradient computations). For Lipschitz, smooth, and strongly convex loss functions, we prove optimal utility with a small gradient complexity for noisy gradient descent algorithms.

## 1   Introduction

Machine learning models leak a significant amount of information about their training data, through their parameters and predictions [21, 18, 5]. Iterative randomized training algorithms can limit this information leakage and bound the differential privacy loss of the learning process [3, 1, 8, 9]. The strength of this certified defense is determined by an *upper bound* on the (Rényi) divergence between the probability distributions of model parameters learned on any pair of neighboring datasets.

The general method to compute the differential privacy bound for gradient perturbation-based learning algorithms is to view the process as a number of (identical) differential privacy mechanisms, and to compute the *composition* of their bounds. However, this over-estimates the privacy loss of the released model [13, 19], and results in a loose differential privacy bound. This is because composition bounds also accounts for the leakage of all intermediate gradient updates, even though only the final model parameters are observable to adversary. Feldman et al. [8, 9] address this issue for the privacy analysis of gradient computations over *one single* training epoch, for smooth and convex loss functions. However, in learning a model over multiple training epochs, such a guarantee is quantitatively similar to composition bounds of privacy amplification by sub-sampling [8]. The open challenge, that we tackle in this paper, is to provide an analysis that can tightly bound the privacy loss of the *released model* after $K$ training epochs, for any $K$.

We present a novel analysis for privacy dynamics of noisy gradient descent with smooth and strongly convex loss functions. We construct a pair of continuous-time Langevin diffusion [20] processes that trace the probability distributions over the model parameters of noisy GD. Subsequently, we derive differential inequalities bounding the *rate* of privacy loss (worst case Rényi divergence between the **coupled stochastic processes** associated with neighboring datasets) throughout the training process.

---

[*]Equal contribution. Alphabetical Order.

35th Conference on Neural Information Processing Systems (NeurIPS 2021).

We then prove an exponentially-fast converging privacy bound for noisy GD: (simplified theorem) Under 1-strongly convex and $\beta$-smooth loss function $\ell(\theta; \mathbf{x})$ with total gradient sensitivity 1, the noisy GD Algorithm 1, with initial parameter vector $\theta_0 \sim \Pi_{\mathcal{C}}(\mathcal{N}(0, 2\sigma^2 \mathbb{I}_d))$ and step size $\eta < \frac{1}{\beta}$, satisfies $(\alpha, \varepsilon)$-Rényi DP with $\varepsilon = O\left(\frac{\alpha}{\sigma^2 n^2}(1 - e^{-\eta \frac{K}{2}})\right)$, where $n$ is the size of the training set.

This guarantee shows that the privacy loss **converges** exponentially in the number of iterations $K$, instead of growing proportionally with $K$ as in the composition-based analysis of the same algorithms. Our bound captures a strong privacy amplification due to the dynamics (and convergence) of differential privacy over the noisy gradient descent algorithm with private internal state.

We analyze the *tightness* of the bound, the *utility* of the randomized algorithm under the computed differential privacy bound, as well as its *gradient complexity* (number of required gradient computations). We prove the **tightness guarantee** for our bound by showing that there exist a loss function and neighboring datasets such that the divergence between corresponding model parameter distributions matches our privacy bound. For Lipschitz, smooth, and strongly convex loss functions, we prove that noisy GD achieves **optimal utility** under differential privacy with an error of order $O(\frac{d}{n^2 \varepsilon^2})$, with a *small gradient complexity* of order $O(n \log(n))$. This improves over the prior utility results for noisy SGD algorithms [3]. Our analysis results in a significantly smaller gradient complexity by a factor of $n/\log(n)$, and a slightly better utility by a factor of $\mathrm{polylog}(n)$.

We anticipate that our work will have a positive societal impact, by paving the way for building accurate and privacy preserving machine learning systems for sensitive personal data.

## 2   Preliminaries on differential privacy

Let $\mathcal{X}$ be the data universe, and a dataset $D$ contain $n$ records from it: $D = (\mathbf{x}_1, \mathbf{x}_2, \cdots, \mathbf{x}_n) \in \mathcal{X}^n$. We refer to a dataset pair $D, D'$ as *neighboring* if they differ in one data record. A measure $\nu$ is said to be absolutely continuous with respect to another measure $\nu'$ on same space (denoted as $\nu \ll \nu'$) if for all measurable set $S$, $\nu(S) = 0$ whenever $\nu'(S) = 0$.

**Definition 2.1** ([17] Rényi differential privacy). *Let $\alpha > 1$. A randomized algorithm $\mathcal{A} : \mathcal{X}^n \to \mathbb{R}^d$ satisfies $(\alpha, \varepsilon)$-Rényi Differential Privacy (RDP), if for any two neighboring datasets $D, D' \in \mathcal{X}^n$, the $\alpha$ Rényi divergence $R_\alpha(\mathcal{A}(D) \| \mathcal{A}(D')) \leq \varepsilon$. For a pair of measures $\nu, \nu'$ over the same space with $\nu \ll \nu'$, $R_\alpha(\nu \| \nu')$ is defined as*

$$R_\alpha(\nu \| \nu') = \frac{1}{\alpha - 1} \log E_\alpha(\nu \| \nu'), \quad where \quad E_\alpha(\nu \| \nu') = \int \left(\frac{d\nu}{d\nu'}\right)^\alpha d\nu'. \tag{1}$$

We refer to $R_\alpha(\mathcal{A}(D) \| \mathcal{A}(D'))$ also as the *Rényi privacy loss* of algorithm $\mathcal{A}$ on datasets $D, D'$. An RDP guarantee can be converted to $(\varepsilon, \delta)$-DP guarantee [17, Proposition 5].

**Definition 2.2** ([22] Rényi information). *Let $\alpha > 1$. For any two measures $\nu, \nu'$ over $\mathbb{R}^d$ with $\mu \ll \nu'$ and corresponding probability density functions $p, p'$, if $\frac{p(\theta)}{p'(\theta)}$ is differentiable, the $\alpha$-Rényi Information of $\nu$ with respect to $\nu'$ is*

$$I_\alpha(\nu \| \nu') = \frac{4}{\alpha^2} \mathop{\mathbb{E}}_{\theta \sim p'} \left[\left\|\nabla \frac{p(\theta)^{\frac{\alpha}{2}}}{p'(\theta)^{\frac{\alpha}{2}}}\right\|_2^2\right] = \mathop{\mathbb{E}}_{\theta \sim p'}\left[\frac{p(\theta)^{\alpha-2}}{p'(\theta)^{\alpha-2}}\left\|\nabla \frac{p(\theta)}{p'(\theta)}\right\|_2^2\right]. \tag{2}$$

See the Appendix B for a comprehensive presentation of preliminaries.

## 3   Privacy analysis of noisy gradient descent

Let $D = (\mathbf{x}_1, \mathbf{x}_2, \cdots, \mathbf{x}_n)$ be a dataset of size $n$ with records taken from a universe $\mathcal{X}$. For a given machine learning algorithm, let $\ell(\theta; \mathbf{x}) : \mathcal{X} \times \mathbb{R}^d \to \mathbb{R}$ be a loss function of a parameter vector $\theta \in \mathcal{C}$ on the data point $\mathbf{x}$, where $\mathcal{C}$ is a closed convex set (can be $\mathbb{R}^d$).

A generic formulation of the optimization problem to learn the model parameters, is in the form of empirical risk minimization (ERM) with the following objective, where $\mathcal{L}_D(\theta)$ is the empirical loss

of the model, with parameter vector $\theta$, on a dataset $D$.

$$\theta^* = \arg\min_{\theta \in \mathcal{C}} \mathcal{L}_D(\theta), \quad \text{where} \quad \mathcal{L}_D(\theta) = \frac{1}{n} \sum_{\mathbf{x} \in D} \ell(\theta; \mathbf{x}). \tag{3}$$

Releasing this optimization output (i.e., $\theta^*$) can leak information about the dataset $D$, hence violating data privacy. To mitigate this risk, there exist randomized algorithms to ensure that the ($\alpha$-Rényi) privacy loss of the ERM algorithm is upper-bounded by $\varepsilon$, i.e., the algorithm satisfies $(\alpha, \varepsilon)$-RDP.

---

**Algorithm 1** $\mathcal{A}_{\text{Noisy-GD}}$: Noisy Gradient Descent

---

**Input:** Dataset $D = (\mathbf{x}_1, \mathbf{x}_2, \cdots, \mathbf{x}_n)$, loss function $\ell(\theta; \mathbf{x})$, closed convex set $\mathcal{C} \subseteq \mathbb{R}^d$, learning rate $\eta$, noise variance $\sigma^2$, initial parameter vector $\theta_0$.
1: **for** $k = 0, 1, \cdots, K-1$ **do**
2:     $g(\theta_k; D) = \sum_{i=1}^{n} \nabla \ell(\theta_k; \mathbf{x}_i)$
3:     $\theta_{k+1} = \Pi_{\mathcal{C}}(\theta_k - \frac{\eta}{n} g(\theta_k; D) + \sqrt{2\eta}\mathcal{N}(0, \sigma^2 \mathbb{I}_d))$
4: Output $\theta_K$

---

In this paper, our objective is to analyze privacy loss of *Noisy Gradient Descent* (Algorithm 1), which is a randomized ERM algorithm. Let $\theta_k, \theta'_k$ be the parameter vectors at the $k$'th iteration of $\mathcal{A}_{\text{Noisy-GD}}$ on neighboring datasets $D$ and $D'$, respectively. We denote by $\Theta_{\eta k}$ and $\Theta'_{\eta k}$ the corresponding random variables that model $\theta_k$ and $\theta'_k$. We abuse notation to also denote their probability distributions by $\Theta_{\eta k}$ and $\Theta'_{\eta k}$. In this paper, our objective is to model and analyze the **dynamics of differential privacy** of this algorithm. More precisely, we focus on the following.

1. Compute an RDP bound (i.e., the worst case Rényi divergence $R_\alpha (\Theta_K \| \Theta'_K)$ between the output distributions of two neighboring datasets) for Algorithm 1, and analyze its tightness.

2. Compute the contribution of each iteration to the privacy loss. As we go from step $k = 1$ to $K$ in Algorithm 1, we investigate how the algorithm's privacy loss changes as it runs the $k$'th iteration (computed as $R_\alpha \left( \Theta_{\eta k} \middle\| \Theta'_{\eta k} \right) - R_\alpha \left( \Theta_{\eta(k-1)} \middle\| \Theta'_{\eta(k-1)} \right)$).

In the end, we aim to provide a RDP bound that is tight, thus facilitating optimal utility [3]. We emphasize that our goal is to construct a theoretical framework for analyzing privacy loss of releasing the output $\theta_K$ of the algorithm, assuming *private* internal states (i.e., $\theta_1, \cdots, \theta_{K-1}$).

### 3.1 Tracing diffusion for Noisy GD

To analyze the privacy loss of Noisy GD, which is a *discrete-time stochastic process*, we first interpolate each discrete update from $\theta_k$ to $\theta_{k+1}$ with a piece-wise continuously differentiable diffusion process. Let $D$ and $D'$ be a pair of arbitrarily chosen neighboring datasets. Given step-size $\eta$ and initial parameter vector $\theta_0 = \theta'_0$, the respective $k$'th discrete updates in Algorithm 1 on neighboring datasets $D$ and $D'$ are

$$\begin{cases} \theta_{k+1} = \Pi_{\mathcal{C}}(\theta_k - \eta \nabla \mathcal{L}_D(\theta_k) + \sqrt{2\eta\sigma^2}\mathbf{Z}_k), \\ \theta'_{k+1} = \Pi_{\mathcal{C}}(\theta'_k - \eta \nabla \mathcal{L}_D(\theta'_k) + \sqrt{2\eta\sigma^2}\mathbf{Z}_k), \end{cases} \quad \text{with} \quad \mathbf{Z}_k \sim \mathcal{N}(0, \mathbb{I}_d). \tag{4}$$

These two discrete jumps can be interpolated with two stochastic processes $\Theta_t$ and $\Theta'_t$ over time $\eta k \leq t \leq \eta(k+1)$ respectively. At the start of each step, $t = \eta k$, the random variables $\Theta_{\eta k}$ and $\Theta'_{\eta k}$ model the distribution of the $\theta_k$ and $\theta'_k$ in the noisy GD processes respectively. During time $\eta k < t < \eta(k+1)$, we model the respective gradient updates on $D$ and $D'$ with the following stochastic processes.

$$\begin{cases} \Theta_t = \Theta_{\eta k} - \eta \cdot U_1(\Theta_{\eta k}) - (t - \eta k) \cdot U_2(\Theta_{\eta k}) + \sqrt{2(t - \eta k)\sigma^2}\mathbf{Z}_k \\ \Theta'_t = \Theta'_{\eta k} - \eta \cdot U_1(\Theta'_{\eta k}) + (t - \eta k) \cdot U_2(\Theta'_{\eta k}) + \sqrt{2(t - \eta k)\sigma^2}\mathbf{Z}_k \end{cases} \tag{5}$$

where the vectors $U_1(\theta) = \frac{1}{2} (\nabla \mathcal{L}_D(\theta) + \nabla \mathcal{L}_{D'}(\theta))$ and $U_2(\theta) = \frac{1}{2} (\nabla \mathcal{L}_D(\theta) - \nabla \mathcal{L}_{D'}(\theta))$ represent the average and difference between gradients on neighboring datasets $D$ and $D'$ respectively.

At the end of step, i.e. at $t \to \eta(k+1)$, we project $\Theta_t$ and $\Theta'_t$ onto convex set $\mathcal{C}$, and obtain

$$\Theta_{\eta(k+1)} = \Pi_{\mathcal{C}} \left( \lim_{t \to \eta(k+1)^-} \Theta_t \right), \Theta'_{\eta(k+1)} = \Pi_{\mathcal{C}} \left( \lim_{t \to \eta(k+1)^-} \Theta'_t \right). \tag{6}$$

By plugging (5) into (6), we compute that the projected random variable $\Theta_{\eta(k+1)}$ and $\Theta'_{\eta(k+1)}$ have the same distributions as the parameters $\theta_{k+1}$ and $\theta'_{k+1}$ at $k+1^{\text{th}}$ step of noisy GD respectively. Repeating the construction for $k = 0, \cdots, K-1$, we define two piece-wise continuous diffusion processes $\{\Theta_t\}_{t \geq 0}$ and $\{\Theta'_t\}_{t \geq 0}$ whose distributions at time $t = \eta k$ are consistent with $\theta_k$ and $\theta'_k$ in the noisy GD processes (4) for any $k \in \{0, \cdots, K-1\}$.

**Definition 3.1** (Coupled tracing diffusions)**.** *Let $\Theta_0 = \Theta'_0$ be two identically distributed random variables. We refer to the stochastic processes $\{\Theta_t\}_{t \geq 0}$ and $\{\Theta'_t\}_{t \geq 0}$ that evolve along diffusion processes (5) in $\eta k < t < \eta(k+1)$ and undergo projection steps (6) at the end of step $t = \eta(k+1)$, as coupled tracing diffusions for noisy GD on neighboring datasets $D, D'$.*

The Rényi divergence $R_\alpha(\Theta_{\eta K} \| \Theta'_{\eta K})$ reflects the Rényi privacy loss of Algorithm 1 with $K$ steps. Conditioned on observing $\theta_k$ and $\theta'_k$, the processes $\{\Theta_t\}_{\eta k < t < \eta(k+1)}$ and $\{\Theta'_t\}_{\eta k < t < \eta(k+1)}$ in (5) are Langevin diffusions along vector fields $-U_2(\theta_k)$ and $U_2(\theta'_k)$ respectively, for duration $\eta$. Therefore, conditioned on observing $\theta_k$ and $\theta'_k$, the diffusion processes in (5) have the following stochastic differential equations (SDEs) respectively.

$$d\Theta_t = -U_2(\theta_k)dt + \sqrt{2\sigma^2}d\mathbf{W}_t, \quad d\Theta'_t = U_2(\theta'_k)dt + \sqrt{2\sigma^2}d\mathbf{W}_t, \tag{7}$$

where $d\mathbf{W}_t \sim \sqrt{dt}\mathcal{N}(0, \mathbb{I}_d)$ describe the Wiener processes on $\mathbb{R}^d$. Therefore, the conditional probability density functions $p_{t|\eta k}(\theta|\theta_k)$ and $p'_{t|\eta k}(\theta|\theta'_k)$ follow the following Fokker-Planck equation. For brevity, we use $p_{t|\eta k}(\theta|\theta_k)$ and $p'_{t|\eta k}(\theta|\theta'_k)$ to represent the conditional probability density function $p(\Theta_t = \theta | \Theta_{\eta k} = \theta_k)$ and $p(\Theta'_t = \theta | \Theta'_{\eta k} = \theta'_k)$ respectively.

$$\begin{cases} \frac{\partial p_{t|\eta k}(\theta|\theta_k)}{\partial t} = \nabla \cdot \left( p_{t|\eta k}(\theta|\theta_k)U_2(\theta_k) \right) + \sigma^2 \Delta p_{t|\eta k}(\theta|\theta_k) \\ \frac{\partial p'_{t|\eta k}(\theta|\theta'_k)}{\partial t} = -\nabla \cdot \left( p'_{t|\eta k}(\theta|\theta'_k)U_2(\theta'_k) \right) + \sigma^2 \Delta p_{t|\eta k}(\theta|\theta'_k) \end{cases} \tag{8}$$

By taking expectations over probability density function $p_{\eta k}(\theta_k)$ or $p'_{\eta k}(\theta'_k)$ on both sides of (8), we obtain the partial differential equation that models the evolution of (unconditioned) probability density function $p_t(\theta)$ and $p'_t(\theta)$ in the coupled tracing diffusions.

**Lemma 1.** *For coupled tracing diffusion processes (5) in time $\eta k < t < \eta(k+1)$, the equivalent Fokker-Planck equations are*

$$\begin{cases} \frac{\partial p_t(\theta)}{\partial t} = \nabla \cdot (p_t(\theta)V_t(\theta)) + \sigma^2 \Delta p_t(\theta) \\ \frac{\partial p'_t(\theta)}{\partial t} = \nabla \cdot (p'_t(\theta)V'_t(\theta)) + \sigma^2 \Delta p'_t(\theta), \end{cases} \tag{9}$$

*where $V_t(\theta) = -V'_t(\theta) = \underset{\theta_k \sim p_{\eta k|t}}{\mathbb{E}} [U_2(\theta_k)|\theta]$ are time-dependent vector fields on $\mathbb{R}^d$, and $U_2(\theta) = \frac{1}{2}[\nabla\mathcal{L}_D(\theta) - \nabla\mathcal{L}_{D'}(\theta)]$ is the difference between gradients on neighboring datasets.*

By this density evolution equation, we model the noisy gradient descent updates with coupled tracing diffusions. The tracing diffusion process is similar to Langevin diffusion. Therefore, we first study the privacy dynamics in coupled tracing (Langevin) diffusions.

## 3.2 Privacy erosion in tracing (Langevin) diffusion

The Rényi divergence (privacy loss) $R_\alpha(\Theta_t \| \Theta'_t)$ between coupled tracing diffusion processes increases over time, as the vector fields $V_t, V'_t$ underlying two processes are different. We refer to this phenomenon as **privacy erosion**. This increase is determined by the amount of change in the probability density functions for coupled tracing diffusions, characterized by the Fokker-Planck equations (9) for diffusions under different vector fields.

Using equation (9), we compute a bound on the rate (partial derivative) of $R_\alpha(\Theta_t \| \Theta'_t)$ over time in the following lemma, to model privacy erosion between two different diffusion processes. We refer to *coupled diffusions* as respective diffusion processes under different vector fields $V_t$ and $V'_t$.

**Lemma 2** (Rate of Rényi privacy loss). *Let $V_t$ and $V_t'$ be two vector fields on $\mathbb{R}^d$ corresponding to a pair of arbitrarily chosen neighboring datasets $D$ and $D'$ with $\max\limits_{\theta \in \mathbb{R}^d} \|V_t(\theta) - V_t'(\theta)\|_2 \leq S_v$ for all $t \geq 0$. Then, for corresponding coupled diffusions $\{\Theta_t\}_{t \geq 0}$ and $\{\Theta_t'\}_{t \geq 0}$ under vector fields $V_t$ and $V_t'$ and noise variance $\sigma^2$, the Rényi privacy loss rate at any $t \geq 0$ is upper bounded by*

$$\frac{\partial R_\alpha\left(\Theta_t \| \Theta_t'\right)}{\partial t} \leq \frac{1}{\gamma} \frac{\alpha S_v^2}{4\sigma^2} - (1 - \gamma)\sigma^2 \alpha \frac{I_\alpha\left(\Theta_t \| \Theta_t'\right)}{E_\alpha\left(\Theta_t \| \Theta_t'\right)}. \tag{10}$$

*where $\gamma > 0$ is a tuning parameter that we can fix arbitrarily according to our need.*

Although this lemma bounds the Rényi privacy loss rate, the term $I_\alpha\left(\Theta_t \| \Theta_t'\right)$ depends on unknown distributions $\Theta_t, \Theta_t'$, and is intractable to compute. Even with explicit expressions for distributions $\Theta_t, \Theta_t'$, the calculation would involve integration in $\mathbb{R}^d$ which is computationally prohibitive for large $d$. Note that, however, the ratio $I_\alpha / E_\alpha$ is always positive by definition. Therefore, the Rényi divergence (privacy loss) rate in (10) is bounded by its first component (a constant) given any fixed $\alpha$.

**Theorem 1** (Linear Rényi divergence bound). *Let $V_t$ and $V_t'$ be two vector fields on $\mathbb{R}^d$, with $\max\limits_{\theta \in \mathbb{R}^d} \|V_t(\theta) - V_t'(\theta)\|_2 \leq S_v$ for all $t \geq 0$. Then, the coupled diffusions under vector fields $V_t$ and $V_t'$ with noise variance $\sigma^2$ for time $T$ has $\alpha$-Rényi divergence bounded by $\varepsilon = \frac{\alpha S_v^2 T}{4\sigma^2}$.*

When the vector fields are $V_t = -\nabla \mathcal{L}_D$ and $V_t' = -\nabla \mathcal{L}_{D'}$, the coupled diffusions follow Langevin diffusion. By definition B.10 of total gradient sensitivity, $\max\limits_{\theta \in \mathbb{R}^d} \|\nabla \mathcal{L}_D(\theta) - \nabla \mathcal{L}_{D'}(\theta)\|_2 \leq \frac{S_g}{n}$. Therefore, this naïve privacy analysis gives linear RDP guarantee for Langevin diffusion, which resembles the moment accountant analysis [1]. However, a tighter bound of Rényi privacy loss is possible with finer control of the ratio $I_\alpha\left(\Theta_t \| \Theta_t'\right) / E_\alpha\left(\Theta_t \| \Theta_t'\right)$, which by definition depends on the likelihood ratio between $\Theta_t$ and $\Theta_t'$, thus is connected with Rényi privacy loss itself. When this ratio grows, the Rényi privacy loss rate decreases, thus slowing down privacy loss accumulation, and leading to tighter privacy bound.

**Controlling Rényi privacy loss rate under isoperimetry** We control the $I_\alpha / E_\alpha$ term in lemma 2 by making an *isoperimetric* assumption known as *log-Sobolev inequality* [2], described as follows.

**Definition 3.2** ([12] Log-Sobolev Inequality ($c$-LSI)). *Distribution of a random variable $\Theta$ on $\mathbb{R}^d$ satisfies* logarithmic Sobolev inequality *with parameter $c > 0$, i.e. it is $c$-LSI, if for all functions $f$ in the function set $\mathcal{F}_\Theta = \{f : \mathbb{R}^d \to \mathbb{R} | \nabla f \text{ is continuous, and } \mathbb{E}(f(\Theta)^2) < \infty\}$, we have*

$$\mathbb{E}[f(\Theta)^2 \log f(\Theta)^2] - \mathbb{E}[f(\Theta)^2] \log \mathbb{E}[f(\Theta)^2] \leq \frac{2}{c} \mathbb{E}[\|\nabla f(\Theta)\|_2^2]. \tag{11}$$

LSI was introduced by Gross [12] as a necessary and sufficient condition for rapid convergence of a diffusion processes. Recently, Vempala and Wibisono [22] showed that this isoperimetry condition is sufficient for rapid convergence of Langevin diffusion in Rényi divergence. Under LSI, they provide the following useful lower bound on $I_\alpha / E_\alpha$ for an arbitrary pair of distributions.

**Lemma 3** ([22] $c$-LSI in terms of Rényi Divergence). *Suppose $\Theta_t, \Theta_t' \in \mathbb{R}^d$ are random variables such that the density ratio between distributions of $\Theta_t$ and $\Theta_t'$ lies in $\mathcal{F}_{\Theta_t'}$. Then for any $\alpha \geq 1$,*

$$R_\alpha\left(\Theta_t \| \Theta_t'\right) + \alpha(\alpha - 1)\frac{\partial R_\alpha\left(\Theta_t \| \Theta_t'\right)}{\partial \alpha} \leq \frac{\alpha^2}{2c} \frac{I_\alpha\left(\Theta_t \| \Theta_t'\right)}{E_\alpha\left(\Theta_t \| \Theta_t'\right)}, \tag{12}$$

*if and only if distribution of $\Theta_t'$ satisfies $c$-LSI.*

Note that $\frac{\partial R_\alpha\left(\Theta_t \| \Theta_t'\right)}{\partial \alpha}$ is always positive, as $R_\alpha\left(\Theta_t \| \Theta_t'\right)$ monotonically increases with $\alpha > 1$ [17]. This lemma shows that $I_\alpha\left(\Theta_t \| \Theta_t'\right) / E_\alpha\left(\Theta_t \| \Theta_t'\right)$ grows monotonically with the Rényi privacy loss $R_\alpha\left(\Theta_t \| \Theta_t'\right)$. By Lemma 2, this implies a throttling privacy loss rate as privacy loss accumulates. Combining Lemma 2 and Lemma 3, we therefore model the **dynamics for Rényi privacy loss under $c$-LSI** with the following PDE, which describes the relation between privacy loss, its changes over time, and its change over Rényi parameter $\alpha$. For brevity, let $R(\alpha, t)$ represent $R_\alpha\left(\Theta_t \| \Theta_t'\right)$.

$$\frac{\partial R(\alpha, t)}{\partial t} \leq \frac{1}{\gamma} \frac{\alpha S_v^2}{4\sigma^2} - 2(1 - \gamma)\sigma^2 c \left[\frac{R(\alpha, t)}{\alpha} + (\alpha - 1)\frac{\partial R(\alpha, t)}{\partial \alpha}\right] \tag{13}$$

The initial privacy loss $R(\alpha, 0) = 0$, as $\Theta_0 = \Theta_0'$. The solution for this PDE increases with time $t \geq 0$, and models the erosion of Rényi privacy loss in coupled tracing diffusions $\Theta_t$ and $\Theta_t'$.

### 3.3 Privacy guarantee for Noisy GD

We now use the privacy dynamics (13) of coupled tracing diffusions to analyze the privacy dynamics for noisy GD. We first bound the difference between the underlying vector fields $V_t$ and $V_t'$ for coupled tracing diffusions for noisy GD on neighboring datasets $D$ and $D'$.

**Lemma 4.** *Let $\ell(\theta; \mathbf{x})$ be a loss function on closed convex set $\mathcal{C}$, with a finite total gradient sensitivity $S_g$. Let $\{\Theta_t\}_{t\geq 0}$ and $\{\Theta_t'\}_{t\geq 0}$ be the coupled tracing diffusions for noisy GD on neighboring datasets $D, D' \in \mathcal{X}^n$, under loss $\ell(\theta; \mathbf{x})$ and noise variance $\sigma^2$. Then the difference between underlying vector fields $V_t$ and $V_t'$ for coupled tracing diffusions is bounded by*

$$\max_{\theta \in \mathbb{R}^d} \|V_t(\theta) - V_t'(\theta)\|_2 \leq \frac{S_g}{n}, \tag{14}$$

*where $V_t(\theta)$ and $V_t'(\theta)$ are time-dependent vector fields on $\mathbb{R}^d$, defined in Lemma 1.*

We then substitute $S_v$ in PDE (13) with $S_g/n$, and compute the following PDE modelling Rényi privacy loss dynamics of tracing diffusion at $\eta k < t < \eta(k+1)$, under $c$-LSI condition.

$$\frac{\partial R(\alpha, t)}{\partial t} \leq \frac{1}{\gamma} \frac{\alpha S_g^2}{4\sigma^2 n^2} - 2(1-\gamma)\sigma^2 c \left[ \frac{R(\alpha, t)}{\alpha} + (\alpha - 1)\frac{\partial R(\alpha, t)}{\partial \alpha} \right] \tag{15}$$

We solve this PDE under $\gamma = \frac{1}{2}$ for each time piece, and combine multiple pieces by seeing projection as privacy-preserving post-processing step. We derive the RDP guarantee for the Noisy GD algorithm.

**Theorem 2** (RDP for noisy GD under $c$-LSI). *Let $\{\Theta_t\}_{t\geq 0}$ and $\{\Theta_t'\}_{t\geq 0}$ be the tracing diffusion for $\mathcal{A}_{\text{Noisy-GD}}$ on neighboring datasets $D$ and $D'$, under noise variance $\sigma^2$ and loss function $\ell(\theta; \mathbf{x})$. Let $\ell(\theta; \mathbf{x})$ be a loss function on closed convex set $\mathcal{C} \subseteq \mathbb{R}^d$, with a finite total gradient sensitivity $S_g$. If for any neighboring datasets $D$ and $D'$, the corresponding coupled tracing diffusions $\Theta_t$ and $\Theta_t'$ satisfy $c$-LSI throughout $0 \leq t \leq \eta K$, then $\mathcal{A}_{\text{Noisy-GD}}$ satisfies $(\alpha, \varepsilon)$ Rényi Differential Privacy for*

$$\varepsilon = \frac{\alpha S_g^2}{2c\sigma^4 n^2}(1 - e^{-\sigma^2 c\eta K}). \tag{16}$$

This theorem offers a strong converging privacy guarantee, on the condition that $c$-LSI is satisfied throughout the Noisy GD process. We then analyze the LSI constant $c$ for given Noisy GD process.

**Isoperimetry constants for noisy GD**   When the loss function is strongly convex and smooth, we prove that tracing diffusion of noisy GD satisfies LSI. This is because the gradient descent update is Lipschitz under smooth loss, and the Gaussian noise preserves LSI, as discussed in Appendix D.3.

**Lemma 5** (LSI for noisy GD). *If loss function $\ell(\theta; \mathbf{x})$ is $\lambda$-strongly convex and $\beta$-smooth over a closed convex set $\mathcal{C}$, the step-size is $\eta < \frac{1}{\beta}$, and initial distribution is $\Theta_0 \sim \Pi_{\mathcal{C}}(\mathcal{N}(0, \frac{2\sigma^2}{\lambda}\mathbb{I}_d))$, then the coupled tracing diffusion processes $\{\Theta_t\}_{t\geq 0}$ and $\{\Theta_t'\}_{t\geq 0}$ for noisy GD on any neighboring datasets $D$ and $D'$ satisfy $c$-LSI for any $t \geq 0$ with $c = \frac{\lambda}{2\sigma^2}$.*

Using the LSI constant proved by this lemma, we immediately prove the following RDP bound for noisy GD on Lipschitz smooth strongly convex loss, as a corollary of Theorem 2.

**Corollary 1** (Privacy Guarantee for noisy GD). *Let $\ell(\theta; \mathbf{x})$ be a $\lambda$-strongly convex, and $\beta$-smooth loss function on closed convex set $\mathcal{C}$, with a finite total gradient sensitivity $S_g$, then the noisy gradient descent algorithm (Algorithm 1) with start parameter $\theta_0 \sim \Pi_{\mathcal{C}}(\mathcal{N}(0, \frac{2\sigma^2}{\lambda}\mathbb{I}_d))$, and step-size $\eta < \frac{1}{\beta}$, satisfies $(\alpha, \varepsilon)$ Rényi Differential Privacy with*

$$\varepsilon = \frac{\alpha S_g^2}{\lambda \sigma^2 n^2}(1 - e^{-\lambda \eta K/2}).$$

This privacy bound has quadratic dependence on the total gradient sensitivity $S_g$, which is upper bounded by $S_g \leq 2L$ for $L$-Lipschitz loss functions. The smoothness condition $\beta$ restricts the step-size and ensures Lipschitz gradient mapping, thus facilitating LSI by Lemma 5. Figure 1 demonstrates how this RDP guarantee for noisy GD converges with the number of iterations $K$. Through y-axis, we show the $\varepsilon$ guaranteed for noisy GD under various Rényi divergence orders $c$ and strong convexity constant $\lambda$. The RDP order $\alpha$ linearly scales the asymptotic guarantee, but does not affect the convergence rate of RDP guarantee. However, the strong convexity parameter $\lambda$ positively affects the asymptotic guarantee as well as the convergence rate; the larger the strong convexity parameter $\lambda$ is, the stronger the asymptotic RDP guarantee and the faster the convergence.

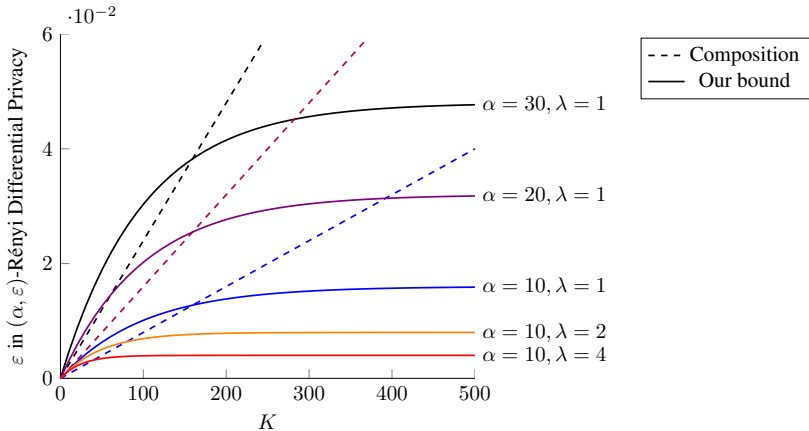

Figure 1: Rényi privacy loss of noisy GD over $K$ iterations, quantified using our DP Dynamics Analysis. We show $\varepsilon$ in the $(\alpha, \varepsilon)$-RDP guarantee derived by Corollary 1 (bold lines), and the Baseline composition analysis (dashed lines). We evaluate under the following setting: RDP order $\alpha \in \{10, 20, 30\}$; $\lambda$-strongly convex loss function with $\lambda \in \{1, 2, 4\}$; $\beta$-smooth loss function with $\beta = 4$; finite $\ell_2$-sensitivity $S_g$ for total gradient with $S_g = 4$; size of the data set $n = 5000$; step-size $\eta = 0.02$; noise standard deviation $\sigma = 0.02$. The expressions for computing the privacy loss are: our analysis: $\varepsilon = \frac{\alpha S_g^2}{\lambda \sigma^2 n^2} \cdot \left(1 - e^{-\lambda \eta K / 2}\right)$; and Baseline composition-based analysis (derived by moment accountant [1] with details in Appendix E): $\varepsilon = \frac{\alpha S_g^2}{4 n^2 \sigma^2} \cdot \eta K$.

## 4 Tightness analysis: a lower bound on privacy loss of noisy GD

Differential privacy guarantees reflect a *bound* on privacy loss on an algorithm; thus, it is very crucial to also have an analysis of their tightness (i.e., how close they are to the exact privacy loss). We prove that our RDP guarantee in Theorem 2 is tight. To this end, we construct an instance of the ERM optimization problem, for which we show that the Rényi privacy loss of the noisy GD algorithm grows at an order matching our guarantee in Theorem 2.

It is very challenging to lower bound the the exact Rényi privacy loss $R_\alpha \left(\Theta_{K\eta} \| \Theta'_{K\eta}\right)$ in general. This might require having an explicit expression for the probability distribution over the last-iterate parameters $\theta_k$. Computing a close-form expression is, however, feasible when the loss gradients are linear. This is due to the fact that, after a sequence of linear transformations and Gaussian noise additions, the parameters follow a Gaussian distribution. Therefore, we construct such an ERM objective, compute the exact privacy loss, and prove the following lower bound.

**Theorem 3** (Lower bound on RDP of $\mathcal{A}_{\text{Noisy-GD}}$). *There exist two neighboring datasets $D, D' \in \mathcal{X}^n$, a start parameter $\theta_0$, and a smooth loss function $\ell(\theta; \mathbf{x})$ on unconstrained convex set $\mathcal{C} = \mathbb{R}^d$, with a finite total gradient sensitivity $S_g$, such that for any step-size $\eta < 1$, noise variance $\sigma^2 > 0$, and iteration $K \in \mathbb{N}$, the privacy loss of $\mathcal{A}_{\text{Noisy-GD}}$ on $D, D'$ is lower-bounded by*

$$R_\alpha \left(\Theta_{\eta K} \| \Theta'_{\eta K}\right) \geq \frac{\alpha S_g^2}{4 \sigma^2 n^2} \left(1 - e^{-\eta K}\right). \tag{17}$$

We prove this lower bound using the $\ell_2$-squared norm loss as ERM objective: $\min_{\theta \in \mathbb{R}^d} \sum_{i=1}^n \frac{\|\theta - \mathbf{x}_i\|_2^2}{n}$. We assume bounded data domain s.t. the gradient has finite sensitivity. With start parameter $\theta_0 = 0^d$, the $k^{\text{th}}$ step parameter $\theta_k$ is distributed as Gaussian with mean $\mu_k = \eta \bar{\mathbf{x}} \sum_{i=0}^{k-1} (1 - \eta)^i$ and variance $\sigma_k^2 = \frac{2\eta\sigma^2}{n^2} \sum_{i=0}^{k-1} (1 - \eta)^{2i}$ in each dimension, where $\bar{x} = \sum_{i=1}^n \mathbf{x}_i / n$ is the empirical dataset mean. We explicitly compute the privacy loss at any step $K$, which is lower bounded by $\frac{\alpha S_g^2}{4\sigma^2 n^2} (1 - e^{-\eta K})$.

Meanwhile, Corollary 1 gives our RDP upper bound $\epsilon = \frac{\alpha S_g^2}{\sigma^2 n^2} \left(1 - e^{-\eta K}\right)$ for this same ERM objective. This upper bound matches the lower bound at every step $K$, up to a small constant of 4.

Moreover, Theorem 2 facilitates a smaller RDP upper bound than Corollary 1 by bounding the LSI constant throughout Noisy GD exactly. For squared-norm loss function, Theorem 2 gives the

following tighter RDP upper-bound for Noisy GD, because all intermediate Gaussian distributions satisfy $c$-LSI with $c = \frac{2-\eta}{2\sigma^2}$, as proved in Appendix E.

**Corollary 2** (RDP guarantee of $\mathcal{A}_{\text{Noisy-GD}}$ on $\ell_2$-norm squared loss). *For any two neighboring datasets $D, D' \in \mathcal{X}^n$, start parameter $\theta_0$, step-size $\eta$, noise variance $\sigma^2$, and $K \in \mathbb{N}$, if the loss function is $\ell_2$-norm squared loss $\ell(\theta; \mathbf{x}) = \frac{1}{2}\|\theta - \mathbf{x}\|_2^2$ on unconstrained convex set $\mathcal{C} = \mathbb{R}^d$, with a finite total gradient sensitivity $S_g$, the privacy loss of $\mathcal{A}_{\text{Noisy-GD}}$ on $D, D'$ is upper-bounded by*

$$R_\alpha\left(\Theta_{\eta K} \| \Theta'_{\eta K}\right) \leq \frac{\alpha S_g^2}{(2-\eta)\sigma^2 n^2}(1 - e^{-\frac{2-\eta}{2}\eta K}). \tag{18}$$

This RDP guarantee converges fast to $O(\frac{\alpha S_g^2}{\sigma^2 n^2})$, which matches the lower bound at every step $K$, up to a constant of $\frac{4}{2-\eta} \approx 2$. This immediately shows tightness of our converging RDP guarantee *throughout* the training process, for a converging noisy GD algorithm. A different approach is to completely ignore the dynamics of differential privacy, and instead analyze privacy *only at the convergence time* (or when the algorithm is near convergence). Wang et al. [25], Minami et al. [16] show that sampling from the Gibbs posterior distribution $\nu(\theta) \propto e^{-\mathcal{L}_D(\theta)/\sigma^2}$ for bounded $\mathcal{L}_D$ satisfies differential privacy. However, sampling exactly from the Gibbs distribution is difficult [4]. Thus, Minami et al. [16], Ganesh and Talwar [11] extend the DP guarantees of Gibbs posterior distribution to gradient-descent based samplers such as Unadjusted Langevin Algorithm (ULA) that can sample from distributions arbitrarily close to Gibbs distribution after a sufficient number of iterations $K$ with extremely small step-size $\eta$. Minami et al. [16] compute the distance to convergence in total variation, and Ganesh and Talwar [11] improve the prior bound by measuring the distance in Rényi divergence (building on the rapid convergence results of Vempala and Wibisono [22]). The latter results in a better gradient complexity $\Omega(nd)$, which however still grows with model dimension $d$. In comparison, our DP guarantees are unaffected by parameter dimension $d$, which in practice can be much larger than the dataset size $n$.

In contrast, composition-based privacy bound grows linearly as training proceeds, as shown in Figure 1. When the number of iterations $K$ is small, however, composition-based bound grows at the same rate with the lower bound, as discussed in Appendix E. Therefore, to conclude whether our RDP guarantee is superior to composition-based bound, we need to understand the number of iterations noisy GD needs, to achieve optimal utility. We discuss this in the following section.

## 5 Utility analysis for noisy gradient descent

The randomness, required for satisfying differential privacy, can adversely affect the utility of the trained model. The standard way to measure the utility of a randomized ERM algorithm (for example, $\mathcal{A}_{\text{Noisy-GD}}$) is to quantify its worst case *excess empirical risk*, which is defined as

$$\max_{D \in \mathcal{X}^n} \mathbb{E}[\mathcal{L}_D(\theta) - \mathcal{L}_D(\theta^*)], \tag{19}$$

where $\theta$ is the output of the randomized algorithm $\mathcal{A}_{\text{Noisy-GD}}$ on $D$, $\theta^*$ is the solution to the standard (no privacy) ERM (3), and the expectation is computed over the randomness of the algorithm.

We provide the *optimal* excess empirical risk (utility) of noisy GD algorithm under $(\alpha, \varepsilon')$-RDP constraint. The notion of *optimality* for utility is defined as the smallest upper-bound for excess empirical risk that can be guaranteed under $(\alpha, \varepsilon')$-RDP constraint by tuning the algorithm's hyper-parameters (such as the noise variance $\sigma^2$ and the number of iterations $K$). We focus here on smooth and strongly convex loss functions with a finite total gradient sensitivity.

**Lemma 6** (Excess empirical risk for smooth and strongly convex loss). *For L-Lipschitz, $\lambda$-strongly convex and $\beta$-smooth loss function $\ell(\theta; \mathbf{x})$ over a closed convex set $\mathcal{C} \subseteq \mathbb{R}^d$, step-size $\eta \leq \frac{\lambda}{2\beta^2}$, and start parameter $\theta_0 \sim \Pi_{\mathcal{C}}(\mathcal{N}(0, \frac{2\sigma^2}{\lambda}\mathbb{I}_d))$, the excess empirical risk of Algorithm 1 is bounded by*

$$\mathbb{E}[\mathcal{L}_D(\theta_K) - \mathcal{L}_D(\theta^*)] \leq \frac{2\beta L^2}{\lambda^2}e^{-\lambda\eta K} + \frac{2\beta d\sigma^2}{\lambda}, \tag{20}$$

*where $\theta^*$ is the minimizer of $\mathcal{L}_D(\theta)$ in the relative interior of convex set $\mathcal{C}$, and $d$ is the dimension of parameter.*

Table 1: Utility comparison with the prior $(\varepsilon, \delta)$-DP ERM algorithms. We assume 1-Lipschitz, $\beta$-smooth and $\lambda$-strongly convex loss. Size of input dataset is $n$, and dimension of parameter vector $\theta$ is $d$. For objective perturbation, we assume $\varepsilon \geq \frac{\beta}{2\lambda}$, and loss is twice differentiable. For our result, we assume $\varepsilon \leq 2\log(1/\delta)$. The lower bound is $\Omega\left(\min\left\{1, \frac{d}{\varepsilon^2 n^2}\right\}\right)$ [3]. We ignore numerical constants and multiplicative dependence on $\log(1/\delta)$.

| | Method | Utility Upper Bound | Gradient complexity |
|---|---|---|---|
| Bassily et al. [3] | Noisy SGD | $O(\frac{d\log^2(n)}{\lambda n^2 \epsilon^2})$ | $n^2$ |
| Wang et al. [23] | DP-SVRG | $O(\frac{d\log(n)}{\lambda n^2 \epsilon^2})$ | $O\left((n + \frac{\beta}{\lambda})\log(\frac{\lambda n^2 \varepsilon^2}{d})\right)$ |
| Zhang et al. [26] | Output Perturbation | $O(\frac{\beta d}{\lambda^2 n^2 \epsilon^2})$ | $O(\frac{\beta}{\lambda} n \log(\frac{n^2 \varepsilon^2}{d}))$ |
| Kifer et al. [14] | Objective Perturbation | $O(\frac{d}{\lambda n^2 \epsilon^2})$ | NA |
| This Paper | Noisy GD | $O(\frac{\beta d}{\lambda^2 n^2 \epsilon^2})$ | $O(\frac{\beta^2}{\lambda^2} n \log\left(\frac{n^2 \varepsilon^2}{d}\right))$ |

This lemma shows decreasing excess empirical risk for noisy GD algorithm under Lipschitz smooth strongly convex loss function as the number of iterations $K$ increases. The utility is determined by $K$ and the noise variance $\sigma^2$, which are constrained under $(\alpha, \varepsilon')$-RDP. Using our tight RDP guarantee in Corollary 1, we prove optimal utility for noisy GD.

**Theorem 4** (Upper bound for $(\alpha, \varepsilon')$-RDP and $(\varepsilon, \delta)$-DP Noisy GD). *For Lipschitz smooth strongly convex loss function $\ell(\theta; \mathbf{x})$ on a bounded closed convex set $\mathcal{C} \subseteq \mathbb{R}^d$, and dataset $D \in \mathcal{X}^n$ of size $n$, if the step-size $\eta = \frac{\lambda}{2\beta^2}$ and the initial parameter $\theta_0 \sim \Pi_{\mathcal{C}}(\mathcal{N}(0, \frac{2\sigma^2}{\lambda}\mathbb{I}_d))$, then the noisy GD Algorithm 1 is $(\alpha, \varepsilon')$-Rényi differentially private, where $\alpha > 1$ and $\varepsilon' > 0$, and satisfies*

$$\mathbb{E}[\mathcal{L}_D(\theta_{K^*}) - \mathcal{L}_D(\theta^*)] = O(\frac{\alpha\beta dL^2}{\varepsilon' \lambda^2 n^2}), \tag{21}$$

*by setting noise variance $\sigma^2 = \frac{4\alpha L^2}{\lambda \varepsilon' n^2}$, and number of updates $K^* = \frac{2\beta^2}{\lambda^2}\log(\frac{n^2 \varepsilon'}{\alpha d})$.*

*Equivalently, for $\varepsilon \leq 2\log(1/\delta)$ and $\delta > 0$, Algorithm 1 is $(\varepsilon, \delta)$-differentially private, and satisfies*

$$\mathbb{E}[\mathcal{L}_D(\theta_{K^*}) - \mathcal{L}_D(\theta^*)] = O(\frac{\beta dL^2 \log(1/\delta)}{\epsilon^2 \lambda^2 n^2}), \tag{22}$$

*by setting noise variance $\sigma^2 = \frac{8L^2(\varepsilon + 2\log(1/\delta))}{\lambda \varepsilon^2 n^2}$, and number of updates $K^* = \frac{2\beta^2}{\lambda^2}\log(\frac{n^2 \varepsilon^2}{4\log(1/\delta)d})$.*

Our algorithm achieves this utility guarantee with $O(\frac{\beta^2}{\lambda^2} n \log\left(\frac{\varepsilon^2 n^2}{d}\right))$ gradient computations of $\nabla\ell(\theta; \mathbf{x})$, which is faster than noisy SGD algorithm [3] with a factor of $n$. However, we additionally assume smoothness for the loss function. Our gradient complexity also matches that of other efficient gradient perturbation and output perturbation methods [23, 26], as shown in Table 1.

This utility matches the following theoretical lower bound in Bassily et al. [3] for the best attainable utility of $(\epsilon, \delta)$-differentially private algorithms on Lipschitz smooth strongly convex loss functions.

**Theorem 5** ([3] Lower bound for $(\varepsilon, \delta)$-DP algorithms). *Let $n, d \in \mathbb{N}$, $\varepsilon > 0$, and $\delta = o(\frac{1}{n})$. For every $(\varepsilon, \delta)$-differentially private algorithm $\mathcal{A}$ (whose output is denoted by $\theta^{priv}$), there is a dataset $D \in \mathcal{X}^n$ such that, with probability at least $1/3$ (over the algorithm random coins), we must have*

$$\mathcal{L}_D(\theta^{priv}) - \mathcal{L}_D(\theta^*) = \Omega\left(\min\left\{1, \frac{d}{\varepsilon^2 n^2}\right\}\right), \tag{23}$$

*where $\theta^*$ minimizes a constructed 1-Lipschitz, 1-strongly convex objective $\mathcal{L}_D(\theta)$ over convex set $\mathcal{C}$.*

Our utility matches this lower bound upto the constant factor $\log(1/\delta)$, when assuming $\frac{\beta}{\lambda^2} = O(1)$. This improves upon the previous gradient perturbation methods [3, 24] by a factor of $\log(n)$, and matches the utility of previously know optimal ERM algorithm for Lipschitz smooth strongly convex loss functions, such as objective perturbation [6, 14] and output perturbation [26].

**Utility gain from tight privacy guarantee**   As shown in Table 1, our utility guarantee for noisy GD is logarithmically better than that for noisy SGD in Bassily et al. [3], although the two algorithms are extremely similar. This is because we use our tight RDP guarantee, while Bassily et al. [3] use a composition-based privacy bound. More specifically, noisy SGD needs $n^2$ iterations to achieve the optimal utility, as shown in Table 1. This number of iterations is large enough for the composition-based privacy bound to grow above our RDP guarantee, thus leaving room for improving privacy utility trade-off, as we further discuss in Appendix F. This concludes that our tight privacy guarantee enables providing a superior privacy-utility trade-off, for Lipschitz, strongly convex, and smooth loss functions.

Our algorithm also has significantly smaller gradient complexity than noisy SGD [3], for strongly convex loss functions, by a factor of $n/\log n$. We use a (moderately large) constant step-size, thus achieving fast convergence to optimal utility. However, noisy SGD [3] uses a decreasing step-size, thus requiring more iterations to reach optimal utility.

## 6   Conclusions

We have developed a novel theoretical framework for analyzing the dynamics of privacy loss for noisy gradient descent algorithms. Our theoretical results show that by hiding the internal state of the training algorithm (over many iterations over the whole data), we can tightly analyze the rate of information leakage throughout training, and derive a bound that is significantly tighter than that of composition-based approaches.

**Future Work.**   Our main result is a tight privacy guarantee for Noisy GD on smooth and strongly convex loss functions. The assumptions are very similar to that of the prior work on privacy amplification by iteration [8], and have obvious advantages in enabling the tightness and utility analysis. However, the remaining open challenge is to extend this analysis to non-smooth and non-convex loss functions, and stochastic gradient updates, which are used notably in deep learning.

## Acknowledgements and Funding

The authors would like to thank Joe P. Chen, Hedong Zhang and anonymous reviewers for helpful discussions on the earlier versions of this paper.

This research is supported in part by Intel Faculty Award (within the www.private-ai.org center), Huawei, Google Faculty Award, VMWare Early Career Faculty Award, and the National Research Foundation, Singapore under its Strategic Capability Research Centres Funding Initiative (any opinions, findings and conclusions or recommendations expressed in this material are those of the authors and do not reflect the views of National Research Foundation, Singapore).

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
