# OpenReview forum: "Differential Privacy Dynamics of Langevin Diffusion and Noisy Gradient Descent"
_NeurIPS.cc/2021/Conference — NeurIPS 2021 Spotlight_

### Official Review · Reviewer_csAH · 2021-07-14

**Rating:** 7
**Confidence:** 3

**Summary:**

The paper studies the dynamics of privacy loss for noisy gradient descent algorithm, whose continuous-time analogue is the Langevin diffusion process. They obtain a tight bound on the Renyi divergence between the pair of probability distributions over parameters of models with neighboring datasets. They prove for smooth and strongly convex loss functions, the privacy loss converges exponentially in the number of iterations, instead of square root in the number of iterations as in composition-based analysis of the same algorithms in the literature. For Lipschitz, smooth and strongly convex loss functions, they prove optimal utility for differential policy algorithms with a small gradient complexity.

**Limitations And Societal Impact:**

The authors have stated social impact at the end of Section 1 and the limitations in the conclusion section at the end of the paper.

**Main Review:**

The paper is well-written, and to the best of my knowledge the results are new. However, I have the following concerns.

(1) The paper is restricted to strongly-convex and smooth loss function, which is not true in general for deep learning applications. I'm wondering whether it will be possible to relax such assumptions. For example, in the literature of Langevin algorithms in the context of sampling and non-convex optimizations, there are studies about convergence results when the target is non-convex; see e.g. Raginsky et al. Non-convex learning via stochastic gradient Langevin dynamics: a nonasymptotic analysis COLT 2017. There are also many more recent references.

(2) Noisy gradient algorithm is a bit restrictive. If you use SGD, the corresponding continuous time SDE is so-called stochastic modified equation in the literature, but I understand that your analysis would fail. Maybe you should mention in the paper the limitations of noisy gradient algorithm.

(3) I find some notations and math presentations a bit sloppy throughout the paper. For example, in Definition 2.1 and Definition 2.2., you wrote that for a pair of distribution p, p', but then what is p(\theta)? Is it the distribution of the probability density function? It seems to me you don't differentiate distribution and probability density function. If you use distribution, then you need to use Radon-Nikodymn derivative for instance. Same thing happens when you talk about conditional distribution in the appendix.

(4) In the appendix, I saw you assume the loss function is B-bounded. That's quite restrictive. Somehow I didn't see that mentioned in the main paper. Where do you need to use this?

(5) In the paragraph after Lemma 3, you used the terminology PDE, but in (12), you have an inequality instead of an equality.

(6) In the sentence after (7), write "(k+1)th step" instead.

(7) In section 6, it should be "an exponential rate".

(8) In reference [7], there shouldn't be "et al." Also, you need to capitalize some words in the references. For example, langevin should be Langevin in [7], [10], [19] etc.

**Time Spent Reviewing:**

2.5

---

> ### Author Response · Authors · 2021-08-10
> **Response to reviewer caAH**
>
>
> **Regarding relaxing the assumption of strongly convex smooth loss functions:** This is an important problem. Some assumptions in our analysis impose technical difficulties, for which the results in the literature for non-convex loss functions **cannot be used in a straight-forward manner** (to the best of our knowledge). We understand these assumptions may limit the application of our analysis, however, we view our current analysis for strongly convex and smooth loss as a first major step towards privacy analysis for black-box noisy GD under **more complex loss functions**. To extend our analysis to the general non-convex non-smooth loss function, we believe the key **technical challenges** are the following.
>   - **Adding appropriate conditions for loss functions:** To our knowledge, obtaining general divergence bounds for distributions under non-convex loss functions is very difficult. Therefore, existing works often make various additional assumptions to facilitate analysis. This includes dissipative condition in Raginsky et al. [b], Hessian Lipschitz condition in Vempala and Wibisono [21], and Boundedness loss condition in Li et al. [14].
>   - **Bounding the LSI constant throughout tracing diffusions:** Although our privacy bound Theorem 2 holds for noisy GD under non-convex loss function, bounding the LSI constant $c$ is another technical challenge. Previously, many results, such as Raginsky et al. [b], prove LSI constant bound for the **stationary Gibbs distribution** of SGLD under non-convex loss. However, our framework requires a **stricter** LSI constant bound **throughout** the tracing diffusion processes, which poses new technical challenges. We are only aware of one such result in Li et al. [14] for SGLD on bounded loss with $\ell_2$-regularization.
>
>
>
> **Regarding the restrictions for extending our analysis to SGD updates:** We answered a similar question in details in response to [Q9] Reviewer wSUc. We acknowledge that the noisy gradient method is more restrictive than SGD, and we will further clarify this limitation in our paper. We will also add the potential approaches to overcome this limitation (as mentioned under [Q9] Reviewer wSUc) in our future work.
>
>
>
> **Regarding notation abuse for distributions and probability density functions:**  We will provide a more precise notations for referring to probility distributions and probability densitiy functions.
>   - For brevity in our submitted paper, we used the probability density function $p,p'$ to represent distributions in Definition 2.1 and Definition 2.2. This is feasible because all distributions during one step of our tracing diffusion {$\Theta_t$}$\_{\eta k<t<\eta (k+1)}$ have full support $\mathbb{R}^d$ and smooth density (due to the convolution with Gaussian noise).
>   - To be more rigorous, we will fix our notation abuse, and use distributions $\mu,\nu$ on $\mathbb{R}^d$ and Radon-Nikodym derivative $\frac{d\mu}{d\nu}$ in Definition 2.1. and Definition 2.2. We will then explain how to compute the Rényi divergence and Rényi information, using probability density(mass) functions $p,p'$ for distributions in our tracing diffusions. In later sections, we will only use notations $p,p'$ to denote the probability density (mass) functions.
>
>
>
> **Regarding the assumption of bounded loss function in the appendix:** We will delete the bounded loss function assumption in our preliminary to avoid confusion. This was an assumption that we had used in an earlier draft of our paper, which we do not assume in the submitted version.
>
>
>
> **Regarding the typos:** We will correct the our inaccurate terminology "PDE", inappropriate article "a", and other typos in our references.
>
> Reference:
> - [b] Raginsky, M., Rakhlin, A., and Telgarsky, M. "Non-convex learning via stochastic gradient Langevin dynamics: a nonasymptotic analysis." COLT, 2017.

---

> > ### Comment · Reviewer_csAH · 2021-08-23
> > **response**
> >
> > I'm satisfied with the authors response. I increased my score. Please make sure you include the discussions on the restrictions of your noisy SGD assumption and the strongly convexity assumption, and fix the notations issues.

---

> > > ### Author Response · Authors · 2021-08-24
> > > **Response to reviewer csAH**
> > >
> > > Thanks for the feedbacks and suggestions. We will make sure to include all the discussions about restrictions of our assumptions in the main text or in the appendix, and we will fix the notation issues for future versions.

---

### Official Review · Reviewer_vsV6 · 2021-07-14

**Rating:** 7
**Confidence:** 3

**Summary:**

This paper investigates the information leakage of noisy gradient descent (GD) algorithms for smooth and strongly convex loss functions. The authors analyze the dynamics of (Renyi) privacy loss during training (over K iterations) when the internal state of the training algorithm is not observable. This leads to an upper bound on the rate of privacy loss of the released model by way of the Renyi divergence between a pair of distributions over model parameters learned on neighboring datasets. The bound turns out to be much better than that obtained using composition-based approaches.


**Limitations And Societal Impact:**

Addressed by the authors. I do not foresee any potential negative societal impact of this work.

**Main Review:**

The paper analyzes the privacy loss dynamics of a noisy version of GD where the iterates are perturbed by isotropic Gaussian noise. I found the paper well-written and the results coherently presented. Also the paper is of topical interest to the NeurIPS community, esp. in relation to differential privacy analysis of noisy versions of GD and its derivatives.

For strongly convex and smooth loss functions, the authors show that noisy GD satisfies an isoperimetric assumption. This allows for a strong guarantee (Theorem 2 and Corollary 1) on the rate of privacy loss that converges exponentially in the number of iterations K. The authors also show that the bound is tight and enjoys a superior privacy-utility tradeoff compared to composition-based approaches.

The assumption of strongly convex and smooth losses limits the practical applicability in deep learning and this has been acknowledged by the authors.


**Time Spent Reviewing:**

3

---

### Official Review · Reviewer_3Yfj · 2021-07-16

**Rating:** 7
**Confidence:** 3

**Summary:**

The paper analyze Differentially Private Gradient Descent (DPGD) for optimizing a function f, which perform gradient descent while adding Gaussian noise to the gradients. The authors consider when f is the sum of loss functions of users in a database, and we want to output a minimizer of f with approximate differential privacy. They analyze the privacy loss and error of DPGD by viewing it as an instance of Langevin Dynamics, which samples from the distribution p with pdf proportional to e^{-f(x)}. The main result of the paper is a bound on the Renyi divergences between two DPGD runs on adjacent databases. The main qualitative improvement of this bound is that the dependence on the number of iterations of DPGD, K, is something like (1-e^{-K}) when the loss functions f are strongly convex and smooth. That is, the divergence bound increases with K, but the asymptotic dependence on K is O(1). In comparison, many past bounds have a small polynomial dependence on K, since they effectively bound the divergence between the full chain of iterates produced by DPGD, rather than the final iterate. In particular, if we took K to be sufficiently large, this would enable sampling from a distribution very close to p with approximate DP. The authors show that indeed, this gets loss that has optimal dependence on dimension, epsilon, and the number of users n.

**Limitations And Societal Impact:**

Yes

**Main Review:**

Overall, I am excited about the results in this paper. The a privacy-preserving Markov chain that has final privacy parameters independent of the number of iterations used is in some sense a "holy grail" of DP MCMC algorithms, and as the authors point out many composition-based analyses in the past are unable to achieve this. Using a Fokker-Planck equation to derive a rate of change for the Renyi divergence that decreases over a time is a very elegant way to get around composition-based analyses, and to my understanding while many analyses using similar ideas appear in the sampling literature, using this type of analysis for privacy is a novel technical contribution. As the authors mention in their submission comments, the requirement that the loss functions be strongly convex and smooth is an impractical requirement, and the use of GD instead of SGD is also somewhat impractical. However, for a theoretically focused paper I would not view these as strong limitations on the results in the paper. For stochastic gradient Langevin dynamics, the theoretical convergence rates even in weaker metrics like KL-divergence are very poorly understood, and most bounds (at least to my knowledge), especially in the non-convex setting, have large polynomial or even exponential dependence on the dimension and desired error. In addition, I'd view a work like this as a first step towards better analyses of SGD for non-convex loss functions in the future, rather than falling short of analyzing them.

In addition, overall I felt that while the paper was technically involved, I was able to come out of reading the paper on the first pass with a good understanding of the overall framework of the authors' approach and what individual steps are needed for their main result. In terms of clarity, the only major comment I have is that it might be nice to state the past results more explicitly in the introduction to better facilitate a comparison with this paper's, but as is the authors do a good job drawing these comparisons in the text. There are also some number of typos in the text, but these can easily be fixed before the conference, and do not really affect the clarify of the result. Lastly, it might be nice to give an interesting example of a sum of loss functions that satisfy the desired assumptions to give some intuition behind the setting the assumptions place us in; it took some time for me to quickly rationalize why having finite total gradient sensitivity did not contradict strong convexity of the loss functions.

In summary, the paper uses a new analysis that cleanly avoids issues past analyses has to get a long-sought-after bound on the privacy loss of Langevin Dynamics, and is organized fairly well. For these reasons, I'd recommend accepting the paper.

**Time Spent Reviewing:**

4

---

> ### Author Response · Authors · 2021-08-10
> **Response to reviewer 3Yfj**
>
> **Regarding more explicit comparsion with past results:** We elaborate on the comparsion between our result and two most relevant works by Vempala and Wibisono [21] and Ganesh and Talwar [10], as follows.
> - Regarding Vempala and Wibisono [21]: We wrote a detailed comparsion between [21] and our paper in terms of technical difficulty and novelty of results in the answer to Reviewer wSUc [Q1].
> - Regarding Ganesh and Talwar [10]: The differential privacy bound for Langevin dynamics in Ganesh and Talwar [10] is also motivated by Vempala and Wibisono[21], and has similar problem setting as our paper. However, their **privacy analysis approach is fundamentally different** from ours, and their **resulting privacy bound** is also **different** from ours.
>   - The privacy analysis approach in [10] is based on weak triangle inequality [10, Fact 7] of Renyi divergence, and the convergence of Langevin dynamics to its stationary distribution in Rényi divergence. As a result, the analysis of [10] is **heavily dependent on the convergence of individual Langevin dynamics process** to its stationary distribution. Meanwhile, our analysis is based on the dynamics of Rényi privacy loss itself and do not necessarily require the convergence of underlying tracing diffusion process to a stationary disribution.
>   - They consider strongly convex smooth loss function and derive privacy bound that is polynomially dependent on the parameter dimension $d$, as described after [10, Theorem]. Meanwhile, our privacy analysis considers strongly convex smooth loss functions that additionally have **finite total gradient sensitivity $S_g$**. As a result, our privacy bound in Theorem 2 is **not dependent on dimension $d$**, but instead depends on $S_g$, which is usually much smaller than polynomial of the dimension $d$.
>
>
>
> **Regarding typos in the text:** We will check the paper carefully and fix the typos.
>
>
> **Regarding adding examples of loss functions:** We will add example loss functions, such as the $\ell_2$ squred loss function, after Corollary 1 to facilitate reading and understanding of our theorem conditions.

---

### Official Review · Reviewer_wSUc · 2021-07-17

**Rating:** 7
**Confidence:** 4

**Summary:**

This paper presents a theoretical framework for analyzing the privacy loss of Noisy Gradient Descent during its execution. Using the Renyi divergence as the differential privacy metric, the authors aims to understand how much information leaks for an optimization function
$\frac{1}{n} \sum_{x\in \mathcal{D}} \ell (\theta; \mathbf{x})  $ and for  the dataset $\mathcal{D}=(\mathbf{x}_1,\cdots,\mathbf{x}_n).$

In order to do that, they interpolate the discrete algorithmic method of "Noisy Gradient Descent" to its stochastic continuous analogue, the famous stochastic differential equation of Focker-Plank Equation. The authors carefully construct this tracing diffusion process among two consecutive steps such that $\theta_k = \Theta_{t=\eta k}$ but $\lim_{t\to \eta (k+1)} = \Theta_{t=\eta(k+1)}$ , where $\theta_k, \Theta_{t=\eta k}$ are the  discrete/continuous parameters for the optimization. This semi-continuous interpolation permits the authors to analyze tightly the information disclosure of the initial method of Noisy Gradient Descent using Gaussian perturbation for the case of
of smooth and strongly convex loss functions.

**Limitations And Societal Impact:**

I don't find any direct societal impact. All Differential Privacy & Adaptive Data Analysis results are $\textit{a-fortiori}$  important for their indirect implications to the society since they are either perscriptive by designing and analyzing new efficiently private algorithms or descriptive analyzing the behaviour and the differential privacy loss of existing algorithms. The mathematical limitations of the paper have been discussed in the main-review section.

**Main Review:**

Originality & Quality:

The way that I review this paper is more like studying a "positional" paper that aims to introduce a novel framework for the analysis of iterative algorithms. Actually, with the way it is written, I found it closer to an Adaptive Data Analysis paper, where we want to find the minimum bits of information that we are able to disclose to the users without violating the Renyi-DP guarrantee for Noisy-GD.
While I have written papers in (Noisy) Gradient-Descent Optimization, Langevin Dynamics and Statistical Differential Privacy, I have never
been exposed in this corner of the literature.

$[Q1]$ However, it seems strongly necessary for me to include a discussion about the technical differencies and the novel parts that authors bring with this submission in comparison with [21] ( (S.Vempala and A. Wibisono's work).

In any other case it seems that they improve quantitatively and qualitatively the private loss analysis for Noisy-GD, our main arsenal in private continuous function optimization. Thus the authors study an important problem and resolve it almost completely. The paper is mathematically rigorous and makes strong theoretical contributions.


Clarity:

Here there are my main objections with this submission. The paper is partially very well-written and partially disappointing.
For example until line 122 the flow guides the reader with good intuition behind the techniques used.
However, already there are some annoying typos, which are repeatitively present in the appendix too.

$[Q2]$ For example Eq (6). Did we lose a minus in front of $\nabla L_D$ based on Eq (5)?

$[Q3]$ Imagine the case of the mean reader who has not been exposed before in Wiener processes/Ito martingale processess.
            How did we derive the Equation between (7) and (8) line 127?

In one spatial dimension ''x'', for an Itô process driven by the standard Wiener process $W_t$ and described by the (SDE): $dX_t = \mu(X_t, t) \,dt + \sigma(X_t, t) \,dW_t$ with drift velocity $\mu(X_t, t)$ and diffusion coefficient $D(X_t, t) = \sigma^2(X_t, t)/2$, the Fokker–Planck equation for the probability density $p(x, t)$ of the random variable $X_t$ is:

$\frac{\partial}{\partial t} p(x, t) = -\frac{\partial}{\partial x}\left[\mu(x, t) p(x, t)\right] + \frac{\partial^2}{\partial x^2}\left[D(x, t) p(x, t)\right].$
Now how did we derive the Eq  between (7) and (8)? Also what is happening with the sign "minus" in front of the gradient again?

$Comment$ Especially in the main text, and if it is possible also in the appendix, numbering all one-line equations only can help the readability of the paper. Physicists and Mathematicians of the previous century were seeking for this convinience. It is really a shame to waste it with no reason

In contrast, Lemma 1 and its proof was well-stated and written correspondingly.

Furthermore, while until section 3.1 the authors had tried to provide intuition for each step, Lemma (2) and the section 3.2 for the privacy erosion misses a lot to keep the discussion in high level.

$[Q4]$ Does the Rate of Renyi privacy loss equation belong to the novel parts of this submission?
             Is it the first time that we derive such a rate for the Langevin diffusion?

Either in positive or negative answer, I believe that it would help again for the sake of readability to explain in a high level why we expect this kind of privacy erosion.

$[Q5a]$ For example, is there any discrete analogue which corresponds to Eq (10) that would help the reader to understand the role of each term in this bound?

$[Q5b]$ Could you explain further the line 172? It seems that from Lemma 3, we can earn some intuition for the relation of $R_a$ and $I_a/E_a$? Could you examplify with numbers why  $I_a/E_a$ would grow monotinically with Renyi privacy loss?

Finally another important question is the tightness of the result.

$[Q6]$ On the one hand both upper and lower bound for Noisy-GD are exponential. However there is a gap in the exponent is that correct? Additionally, I did not achieve to understand the figure 1 and especially the baseline composition analysis. I would like to ask the authors to describe in details what this baseline analysis is trying to show as far as this Gap concerns.


In the appendix, I will admit that there was a significant effort to be polished and they cover a lot of background to make the paper self-contained. However there are some points that I would like a more rigorous explanation.

$[A1]$ For the Eq (40) and Eq (41) could you provide the exact statement and possibly the proof of them. Intuitively, I understand obviously the claims (in one dimension they are similar with $\int_{[0,\infty]}x^2 e^{-x}= \int_{[0,\infty]} e^{-x}$). Nevertheless, since these integration rules are used for more complex scenarios, it would be better to explain rigorously phrases like "the gradient decays sufficiently fast at infinity". Additionally, it would be better to present in a short lemma why these assumptions are satisfied in the current setting

$[A2]$ In line 535, in order to exchange the integration and derivative, we have to use some measure theory argument. Since there is no clarity about the quality of the function $(p_t/p_t^\prime)^{a-1}$, could you make this exchange a bit more rigorous?

$[A3]$ For the sake of readability, I would introduce a fact before the proof of lemma 5, to explain why Noisy-GD can be written as a convolution of a pushover distribution and the gaussian noise.

$[A4]$ Finally in 714 (second equation) and 722 line, did you apply KKT conditions? If yes, that means that the constraint set does not include corners (can not be a simplex). Is that correct? I would like a more rigorous explanation about the optimality conditions of $\theta^*$ and the assumptions for the properties of $C$
 like they provide a proof of extension lemma adapted to their setting.


Significance:
In general, after re-reading twice the paper and its appendix, I can say certainly that this is a complete theory paper: It addresses a fundamental problem, analyzes a state-of-art algorithm, and tries to proves the algorithm's privacy behaviour optimally. It tries to describe a high-level technique for other kind of algorithms too which has an independent interest. This is the main reason that my score belongs to the positive side. I will wait the discussion and the response of the authors to my questions to re-examine again my perception for this submission

Some extra questions that I would like to ask the authors?

$[Q7 ]$ Are there other type of noises that your analysis could handle for free or with small amount of effort (like a uniform, any symmetric, log-concave distributions)?

$[Q8 ]$ Where in the proof was used crucially the smoothness and strong convexity of loss function?

$[ Q9 ]$ Multiple theoretical results prove that SGD generalizes better than GD. Is there any straightforward implication about the privacy erosion of the methods? Additionally, what would be the corresponding SDE for the case of SGD? I am asking that because in a significant amount of literature langevin diffusion is used as the continuous analogue of SGD and not GD.



**Time Spent Reviewing:**

7 hours

---

> ### Author Response · Authors · 2021-08-10
> **Response 4 to reviewer wSUc (Q8, Q9, and additional references)**
>
> **[Q8]** There are two places where we require strong convexity and smoothness in our submitted paper.
> - We crucially require strong convexity and smoothness for **proving the LSI constant bound** throughout the tracing diffusion for noisy GD, in Lemma 5.
>   - Strong convexity and smoothness together ensure the contractivity of the gradient descent mapping $\theta-\eta\nabla L_D(\theta)$ over convex set $\mathcal{C}$. Because of this contractivity, the LSI constant bound of the mapped distribution increases after the gradient descent mapping. Subsequently, the additive Gaussian noise to the mapped distribution decreases its LSI constant bound. These together make it possible that, after one gradient descent update with noise, the LSI constant of parameter distribution remains the same.
> - The submitted version of our paper also requires smoothness to bound the privacy loss of the tracing diffusions for noisy GD under $c$-LSI, as shown in Theorem 2. However, since submission, we have found that by constructing new tracing diffusions, **smoothness is not needed for Theorem 2**. We can provide a new proof for Theorem 2 without assuming smooth loss functions, as follows.
>   - **New coupled tracing diffusion for noisy gradient descent:** We can construct two new tracing diffusion processes {$\Theta_t$}$\_{\eta k\leq t\leq \eta (k+1)}$ and {$\Theta_t'$}$\_{\eta k\leq t\leq\eta (k+1)}$ for the $k$th noisy GD update as the following.
>     - At the start $t=k\eta$ of the $k$th step: $\Theta_{k\eta}\leftarrow\theta_k, \Theta_{k\eta}'\leftarrow\theta_k'$.
>     - During $k\eta < t< (k+1)\eta$ of the $k$th step:
>       - $\Theta_t = \Theta_{k\eta} -\frac{\eta}{2}[\nabla L_D(\Theta_{k\eta}) + \nabla L_{D'}(\Theta_{k\eta})] -\frac{t-\eta k}{2}[\nabla L_D(\Theta_{k\eta}) -\nabla L_{D'}(\Theta_{k\eta})] +\sqrt{2(t-\eta k)\sigma^2}Z_k;$
>       - $\Theta_t'=\Theta_{k\eta}' -\frac{\eta}{2}[\nabla L_D(\Theta_{k\eta}) + \nabla L_{D'}(\Theta_{k\eta})] -\frac{t-\eta k}{2}[\nabla L_{D'}(\Theta_{k\eta}')-\nabla L_{D}(\Theta_{k\eta}')] +\sqrt{2(t-\eta k)\sigma^2}Z_k';$
>       - where $Z_k,Z_k'$ are i.i.d $d$-dimensional Gaussian random variables.
>     - At the end $(k+1)\eta$ of the $k$-th step: we do a projection and obtain
>       - $\Theta_{(k+1)\eta}=\lim_{t\rightarrow (k+1)\eta, t<(k+1)\eta} \Pi_{\mathcal{C}}(\Theta_{t});$
>       - $\Theta_{(k+1)\eta}'=\lim_{t\rightarrow (k+1)\eta, t<(k+1)\eta} \Pi_{\mathcal{C}}(\Theta_{t}');$
>       - Comparing with the $k$th noisy GD update in Algorithm 1, we can prove that $\Theta_{(k+1)\eta},\Theta_{(k+1)\eta}'$ recover the distributions for $\theta_{k+1}$ and $\theta_{k+1}'$, similarly to our submitted paper.
>   - **Bounding process difference for the new tracing diffusion:** We can derive an alternative for Lemma 4 **without** assuming smooth loss functions, as follows.
>     - {$\Theta_t$}$\_{\eta k<t<\eta (k+1)}$ can be seen as Langevin diffusion over vector field $\frac{1}{2} \mathbb{E} [\nabla L_D(\Theta_{\eta k})-\nabla L_{D'}(\Theta_{\eta k})|\Theta_t=\theta]$, and {$\Theta_t$}$\_{\eta k<t<\eta (k+1)}$ can be seen as Langevin diffusion over vector field $\frac{1}{2} \mathbb{E}[ \nabla L_{D'}(\Theta_{\eta k}') - \nabla L_{D}(\Theta_{\eta k}') | \Theta_t'=\theta ]$.
>     - By definition of total gradient sensitivity, we bound the magnitude of both vector fields by $\frac{S_g}{2n}$ in $\ell_2$-norm.
>     - By triangle inequality, we bound the difference between vector fields underlying two tracing diffusions $\Theta_t$ and $\Theta_t'$ by $\frac{S_g}{n}$.
>   - **Bounding LSI constant for the new tracing diffusion:** We can prove an alternative for Lemma 5. Similarly, we write the tracing diffusion $\Theta_t$ as a convolution of a $T$-pushover distribution and the Gaussian noise $\mathcal{N}( 0 , 2(t-\eta k)\sigma^2\mathbb{I}\_d )$, where the mapping $T(\theta) = \theta-\frac{\eta}{2}[\nabla L_D(\theta)+\nabla L_{D'}(\theta)]-\frac{t-\eta k}{2}[\nabla L_D(\theta)-\nabla L_D(\theta)]$. We compute the Jacobian of the mapping to be $J\_T(\theta) = \mathbb{I}\_d - \frac{\eta + (t-\eta k)}{2} \nabla^2 L\_D(\theta) - \frac{\eta - (t-\eta k)}{2} \nabla^2 L\_{D'}(\theta)$. We then prove the LSI constant bound $c = \frac{\lambda}{2\sigma^2}$ for tracing diffusion $\Theta_t$ by induction.
>     - Base: $\Theta_{\eta k}$ satisfies $c$-LSI with $c=\frac{\lambda}{2\sigma^2}$.
>     - Induction: we prove that for any $\eta k<t<\eta(k+1)$, $\Theta_t$ satisfies $c$-LSI with $c=\frac{\lambda}{2\sigma^2}$.
>       - By smoothness and strong convexity, the characteristic values of $J_T(\theta)$ falls in $(1-\eta\beta,1-\eta\lambda)$. Therefore $T$ is $(1-\eta\lambda)$-Lipschitz, and $T(\Theta_{\eta k})$ satisfies $c'$-LSI with $c' = \frac{c}{(1-\eta\lambda)^2}$, by Lemma 7.
>       - By convolution between $T(\Theta_{\eta k})$ and Gaussian noise $\mathcal{N}(0,2(t-\eta k)\sigma^2\mathbb{I}\_d)$, we prove that $\Theta_t$ satisfies $c''$-LSI with $c'' = (\frac{1}{c'}+2(t-\eta k)\sigma^2)^{-1}$, by Lemma 8.
>          - Taking $c = \frac{\lambda}{2\sigma^2}$ and $c' = \frac{c}{(1-\eta\lambda)^2}$ into $c''$, we obtain $c'' = \frac{c}{(1-\eta\lambda)^2 + 2(t-\eta k)\sigma^2 c} \geq \frac{c}{1-\eta\lambda+2\eta \sigma^2 c} = \frac{\lambda}{2\sigma^2}$.
>   - **New privacy bound under LSI _without_ assuming smoothness:**
>     - Using the above new process difference bound, we can prove the following new alternative partial differential inequality to replace Eq (14) for privacy erosion. $$\frac{\partial R(\alpha, t)}{\partial t}\leq \frac{1}{\gamma}\frac{\alpha S_g^2}{4\sigma^2n^2} - 2(1-\gamma)\sigma^2c\left[ \frac{R(\alpha,t)}{\alpha} + (\alpha-1) \frac{\partial R(\alpha,t)}{\partial \alpha} \right]$$
>     - Using this new partial differential inequality for privacy erosion, we can prove a new alternative privacy bound to Theorem 2 as $\epsilon = \frac{\alpha S_g^2}{2c\sigma^4 n^2} (1 - e^{-\sigma^2c\eta K})$, **without assuming smooth loss functions**, by following the proof idea for Theorem 2.
>     - Finally, using the new LSI constant bound and the alternative for theorem 2, we can also prove an alternative of Corollary 1 for the privacy bound of noisy GD under strongly convex smooth loss function $\epsilon = \frac{\alpha S_g^2}{\lambda\sigma^2 n^2} (1 - e^{-\lambda\eta K/2})$, which **removes the $\frac{1}{(1-\eta\beta)^2}$ factor**.
>
>
>
>
> **[Q9]** Extending our privacy analysis for (noisy) SGD updates would be an important future work, and there are multiple technical challenges to overcome.
>   - The more achievable way to extend our privacy analysis to noisy SGD lies in the **privacy amplification by sampling** literature, e.g. Mironov et al. [a], which prove that the privacy loss for one SGD update is smaller than the privacy loss of a full-batched GD update by squared sampling ratio. However, our analysis requires quantifying the privacy amplification of multiple SGD updates with iterative re-sampling, which is technically more involved and requires new results.
>   - A more difficult way is to **analyze the privacy erosion for the new (noisy) SGD process**, which poses the following technical challenges.
>     - **Variety of methods to sample mini-batches from the dataset:** Various sampling methods, such as sampling _without replacement_, sampling _with replacement_, _Poisson_ sampling, and the most commonly used _shuffle than partition_, result in different processes during (noisy) SGD update. They often requiring separate privacy analyses.
>     - **Deriving the _exact_ SDE for (noisy) SGD updates:** To the best of our knowledge, the existing literature which uses underlying continuous diffusion processes to trace the (noisy) SGD updates, e.g. in Raginsky et al. [a] and Li et al. [14], usually has approximation error that depends on the discretization step-size $\eta$. Meanwhile, our current privacy analysis would require an underlying diffusion process that **exactly** interpolates the discrete SGD updates, and therefore cannot use these approximate SDEs straightforwardly. To extend our analysis to (noisy) SGD, we either need the **exact** underlying SDE for noisy SGD, or a **new** privacy erosion analysis technique based on **approximate SDE**, which both require more technical investigations.
>     - **Bounding LSI constant throughout tracing diffusion**. Our privacy bound relies on the LSI constant throughout the tracing diffusion process. Meanwhile, previous papers, such as Raginsky et al. [a], only require and prove LSI constant bounds for the stationary Gibbs distribution of SGLD. To our knowledge, for general intermediate distribution in SGLD, it is difficult to prove a (tight) bound on the LSI constant $c$. We are only aware of one such result in Li et al. [14] for SGLD on bounded loss with $\ell_2$-regularization.
>
>
> References:
> - [a] Mironov I., Talwar K., and Zhang L. "R\'enyi Differential Privacy of the Sampled Gaussian Mechanism." arXiv preprint 2019.
> - [b] Raginsky M., Rakhlin A., and Telgarsky M. "Non-convex learning via stochastic gradient Langevin dynamics: a nonasymptotic analysis." COLT, 2017.
> - [c] Olla S, Varadhan S R S, Yau H T. Hydrodynamical limit for a Hamiltonian system with weak noise. Communications in mathematical physics, 1993.
> - [d] Yau H T. Relative entropy and hydrodynamics of Ginzburg-Landau models. Letters in Mathematical Physics, 1991.
> - [e] Timoney R M. Chapter 4: The dominated convergence theorem and applications, MA2224, Lebegue integral, 2017.
> - [f] Boyd, Stephen, Stephen P. Boyd, and Lieven Vandenberghe. Convex optimization. Cambridge university press, 2004.

---

> > ### Comment · Reviewer_wSUc · 2021-08-22
> > **Response to all comments of rebuttal phase**
> >
> > Personally, I am more than satisfied with the answers of the reviewer.
> > I would increase also my score about the current submission. However, I would like to ask from the authors to include all this discussion (and the intuition behind the terms in the main text or in the appendix). Even in the case that there is no space enough in the main text,It would be more than useful for the readability and the future application of  this submission to include them in the appendix in the specific sections as a prelude or footnotes.

---

> > > ### Author Response · Authors · 2021-08-23
> > > **Response to comments of Reviewer wSUc**
> > >
> > > Thanks for the feedback and suggestion, we will include all the discussions in the main text or in the appendix for future versions.

---

> ### Author Response · Authors · 2021-08-10
> **Response 3 to reviewer wSUc (A1, A2, A3, A4, Q7)**
>
>
> **[A1]** Identitiy (41) comes from **applying integration by parts**. When $f: U \rightarrow \mathbb R$ and $g: U \rightarrow \mathbb R$ are two scalar functions defined on some region $U \in \mathbb R^d$ such that $f$ is *once continuously differentiable* and $g$ is *twice continuously differentiable*, Green's first identity says that:
> $$
>   \int_U \left\langle\nabla f, \nabla g \right\rangle dV = \oint_{\partial U} f \nabla g \cdot dS - \int_U f \Delta g\ dV.
> $$
>   The identity in Eq. (41) holds when the surface integral term is zero.
>
>   In the paper, we invoke Eq. (40) and (41) to simplify $F_2$ and $F_1$ respectively in the proof of Lemma 2. In these invocations, the four surface integrals that should approach zero are:
> $$\lim_{r\rightarrow \infty}\oint_{\partial B_r} \left(\frac{p_t}{p_t'}\right)^{\alpha-1} \nabla p_t \cdot dS = 0, $$
> $$ \lim_{r\rightarrow \infty}\oint_{\partial B_r} \left(\frac{p_t}{p_t'}\right)^{\alpha} \nabla p_t' \cdot dS =0, $$
> $$ \lim_{r\rightarrow \infty}\oint_{\partial B_r}\left(\frac{p_t}{p_t'}\right)^{\alpha-1} p_t \nabla \mathcal{L}\_D \cdot dS =0, $$
> $$ \lim_{r\rightarrow \infty} \oint_{\partial B_r} \left(\frac{p_t}{p_t'}\right)^{\alpha-1} p_t \nabla \mathcal{L}\_{D'}\cdot dS = 0,$$
>   where $B_r$ is a ball of radius $r$. From Cauchy-Schwartz inequality, and standard vector algebra, the sufficient condition for the above to hold are:
> $$
>   \lim_{r\rightarrow \infty}\oint_{\partial B_r} \Vert \nabla p_t \Vert^2 \hat n \cdot dS = 0, \quad and \quad \lim_{r\rightarrow \infty} \oint_{\partial B_r} p_t^2 \hat n \cdot dS = 0,
> $$
>   where $\hat n$ is a unit vector perpendicular to surface $dS$. We will add a short lemma in the appendix to prove these boundary conditions for $p_t$ describing our tracing diffusion {$\Theta_t$}$\_{\eta k<t< \eta(k+1)}$}. We will use the fact that $p_t$ is the convolution of a projected distribution and Gaussian distribution.
>
>
> **[A2]** We clarify this exchange of derivative and integration more rigorously, by the measure theory statement for the Leibniz rule (e.g. Theorem 4.4.3 in Timoney et al. [e]). We translate the necessary conditions into our setting as follows.
> - For any $\eta k<t<\eta (k+1)$, the function $(\frac{p_t}{p_t'})^{\alpha-1}$ is Lebesgue-integrable (with respect to measure with density $p_t'$)
> - For any $\eta k<t<\eta (k+1)$, and for almost all $\theta\in\mathbb{R}^d$, the derivative of the function with regard to time $\frac{\partial}{\partial t}(\frac{p_t}{p_t'})^{\alpha-1}$ exists.
> - For any $\eta k<t_0<\eta (k+1)$, there exists a small interval $[t_0, t_0+h_0], h_0>0$, such that the derivative of the function with regard to time $|\frac{\partial}{\partial t}(\frac{p_t}{p_t'})^{\alpha-1}|\leq g(\theta), \forall t\in [t_0,t_0+h_0]$ is bounded by a Lebesgue integrable function $g(\theta)$ (with respect to measure with density $p_t'$) almost everywhere over $\theta\in\mathbb{R}^d$.
>
> We will add a short lemma in the appendix to prove these necessary conditions, by using the following properties of probability density functions $p_t,p_t'$.
>   - $p_t$ and $p_t'$ have the same support, and their Renyi divergence is well-defined.
>   - The distributions of tracing diffusion {$\Theta_t$}$\_{k\eta<t<(k+1)\eta}$, {$\Theta_t'$}$\_{k\eta<t<(k+1)\eta}$ have full support and smooth densities $p_t,p_t'$ (due to convolution with Gaussian noise).
>   - The evolutions of $p_t$ and $p_t'$ with regard to $t$ satisfy the Fokker-planck equation.
>
>
> **[A3]** We will add this description before the proof of Lemma 5.
>
>
>
> **[A4]** We did not use the KKT condition in line 714 (second equation) or line 722. Instead, we used one assumption about the relative position of the optimum $\theta^\*=\arg\min_{\theta\in\mathcal{C}}L_D(\theta)$ in the convex set $\mathcal{C}$. We missed explicitly saying that we made this assumption in the theorem body of Lemma 6, and we will add it in future versions.
>
>   - **The assumption that we missed explicitly saying we made:** In our proof, we used one assumption that we did not explicitly state in Lemma 6: $\theta^\*=\arg\min_{\theta\in\mathcal{C}}L_D(\theta)$ is in the relative interior $\mathbf{relint}(\mathcal{C})$ (defined in section 2.1.3 of Boyd and Vandenberghe[f]) of closed convex set $\mathcal{C}$, i.e. $\theta^\*\in \mathbf{relint}(\mathcal{C})$.
>   - **Loss function property:** The loss function $L_D(\theta)$ is continously differentiable, $\lambda$-strongly convex, $\beta$-smooth and $L$-Lipschitz on a (bounded) closed convex set $\mathcal{C}\subseteq \mathbb{R}^d$.
>   - **Convex set property:** The convex set $\mathcal{C}\subseteq\mathbb{R}^d$ is closed. It can be a convex set with corners, such as probability simplex.
>   - **Rigorous proof for the optimality condition $\langle \nabla L_D(\theta^\*), \theta_k-\theta^\*\rangle=0$ we used in Line 714 and Line 722:**
>     - We construct the function $L(r):\mathbb{R}\rightarrow \mathbb{R}, r \mapsto L_D(\theta^\* + r\cdot (\theta_k-\theta^\*))$ for any $\theta_k$ and the optimum $\theta^\* = \arg\min_{\theta \in \mathcal{C}} L(\theta)$ over the convex set.
>     - Because $\theta^\*\in \mathbf{relint}(\mathcal{C})$, we prove that there exists a small enough $r^\*>0$, such that for any real number $r$ which satisfies $-r^\*\leq r\leq r^\*$, we have $\theta^\* + r (\theta_k-\theta^\*) = (1-r)\theta^\* + r\theta_k \in \textbf{aff}(\mathcal{C})$.
>     - Combining the above property and the optimality of $\theta^\*=\arg\min_{\theta\in\mathcal{C}}L_D(\theta)$, we prove $L_D(\theta^\* + r \cdot (\theta_k-\theta^\*)) \geq L_D(\theta^\*)$ for any $r \in [-r^\*,r^\*]$. Therefore $L(r) \geq L(0)$ for any $r\in[-r^\*,r^\*]$.
>     - Because $L_{D}(\theta)$ is continuously differentiable over $\mathcal{C}$, we prove that $L'(r)$ is continous over $[-r^\*,r^\*]$. Because $0$ is the optimum of $L(r)$ over $[-r^\*,r^\*]$, and that $L'(r)$ is continous over $[-r^\*,r^\*]$, we prove $L'(0) = \langle \nabla L_D(\theta^\*), \theta_k-\theta^\* \rangle = 0$, by mean value theorem for continous function $L'(r)$.
>   - Finally, we want to comment on the **reasonability of our assumption for strongly convex optimization**.
>     - Strongly convex loss function has one unique optimum over the convex set $\mathcal{C}$. Therefore it may be natural to require that this unique optimum is in the relative interior of our constructed convex set $\mathcal{C}$. For example, we see similar assumptions in Theorem 2 of Zhang et al.[25], which assumes the global optimum $\lVert\hat{w}\rVert \leq D$ to be in the convex set $\{w:\lVert w\rVert\leq 2D\}$.
>
>
>
> **[Q7]** We do not see a straightforward extension of our analysis to other noise distributions. The noise distribution in our analysis is restricted to Gaussians because of the following two reasons:
>
> - The increments in a Wiener process is a Gaussian: $W_{t+u} - W_t$ is normally distributed with mean $0$ and variance $u$, i.e. $W_{t+u} - W_t \sim \mathcal N(0, u\mathbb I)$. This allows us to model the noisy gradient descent step as a diffusion process in Eq. (5) with a constant diffusion coefficient $D(\Theta_t, t) = \sigma^2 \mathbb I$. Our simplified Fokker-Planck Eq. (8) is a result of the diffusion coefficient being constant as otherwise, we can't take out $\sigma^2$ outside of the Laplacian operator $\Delta$ (as described in the answer of Q3). When the diffusion coefficient is not constant, multiple steps in our proof of Lemma 2 will be affected.
> - For proving that LSI holds for the tracing diffusion in Lemma 5, we rely on the property that convolving a distribution with Gaussian (i.e. adding Gaussian noise to the gradients) still satisfy LSI (Lemma 8). For uniform, arbitrary symmetric or arbitrary log-concave distributions, such a property might not hold.

---

> ### Author Response · Authors · 2021-08-10
> **Response 2 to reviewer wSUc (Q5a, Q5b, Q6)**
>
> **[Q5a]** We are not aware of any discrete analogue of our bound for the rate of privacy loss in Eq (10) of Lemma 2. We explain the terms of our bound for the rate of privacy loss in Lemma 2 as the following.
> - **$\frac{\alpha S_g^2}{4\sigma^2n^2}$:** This is the first term on the right-hand side of Eq (10). It quantifies the worst-case privacy loss due to one noisy gradient update in noisy GD Algorithm 1, which we explain as follows.
>   - **$\frac{S_g}{n}$** is the sensitivity of average loss gradient $L_D(\theta)$ over two neighboring datasets $D,D'$. The larger $S_g$ is, the further apart the parameters $\theta$ and $\theta'$ after the gradient descent updates on two neighboring dataset $D,D'$ could be, where $\theta=\theta_0-\eta\nabla L_D(\theta_0)$ and $\theta'=\theta_0-\eta L_{D'}(\theta_0)$.
>   - **$\sigma^2$** is the variance of Gaussian noise. Because additive noise shrinks the expected trajectory difference between $\theta$ and $\theta'$ in noisy GD updates, the larger $\sigma^2$ is, the more indistinguishable the distributions of the sum of $\theta,\theta'$, and Gaussian noise will be, therefore the smaller the privacy loss (Rényi divergence between end distributions) will be.
> - **$\frac{I_{\alpha}(\Theta_t\lVert\Theta_t')}{E_{\alpha}(\Theta_t\lVert\Theta_t')}$:** This term is the second term in the right hand side of Eq (10), which originates from the derivative of $p_t,p_t'$ with regard to time $t$. To obtain the expression $I_{\alpha}/E_{\alpha}$, we are using the Fokker Planck equation to replace the terms related to $\frac{\partial p_t}{\partial t}, \frac{\partial p_t'}{\partial t}$ with terms determined by the gradient and Laplacian of $p_t,p_t'$ over $\theta$.
>   - The term $I_{\alpha}$ is the **Rényi information** defined in Definition 2.2., which equals $\mathbb{E}\_{\theta\sim p_t'}\left[\left\lVert \nabla \log \frac{p_t(\theta)}{p_t'(\theta)} \right\rVert_2^2 \left(\frac{p_t(\theta)}{p_t'(\theta)}\right)^\alpha\right]$.
>   - The term $E_{\alpha}$ is the **moment of likelihood ratio** defined in Definition 2.1., which equals $\mathbb{E}\_{\theta\sim p_t'}\left[\left(\frac{p_t(\theta)}{p_t'(\theta)}\right)^{\alpha}\right]$.
>   - These two terms differ by a **multiplicative ratio $\left\lVert \nabla \log \frac{p_t(\theta)}{p_t'(\theta)} \right\rVert_2^2$** for their quantities inside expectation.
>     - This ratio characterizes the variation of log likelihood ratio function across $\theta$, where $\theta$ is taken from distribution $p_t'$. This is intuitive in the one dimensional case, because $\int_{\theta_1}^{\theta_2}\nabla \log \frac{p_t(\theta)}{p_t'(\theta)} d\theta = \log \frac{p_t(\theta_2)}{p_t'(\theta_2)} - \log \frac{p_t(\theta_1)}{p_t'(\theta_2)}$.
>     - Meanwhile since $p_t(\theta),p_t(\theta)'$ are continuous and $\int p_t(\theta)d\theta=\int p_t'(\theta)d\theta=1$, by mean value theorem, there exists $\tilde{\theta}\in\mathbb{R}^d$ such that the log likelihood ratio $\log \frac{p_t(\tilde{\theta})}{p_t'(\tilde{\theta})}$ is zero. Therefore the variation of log likelihood ratio across $\theta$ implicitly increases the largest log likelihood ratio $\max_{\theta\in\mathbb{R}^d}\left[\log(\frac{p_t(\theta)}{p_t(\theta)}) - \log(\frac{p_t(\tilde{\theta})}{p_t(\tilde{\theta})})\right] = \max_{\theta\in\mathbb{R}^d}\left[\log(\frac{p_t(\theta)}{p_t(\theta)})\right]$ across $\theta$ , which reflects the Rényi privacy loss $R_{\alpha}$.
>     - As a result, intuitively, the larger the Rényi privacy loss $R_{\alpha}$ is, the larger the variation of log likelihood ratio across $\theta$ will be, and therefore the larger the term $\frac{I_{\alpha}(\Theta_t\lVert\Theta_t')}{E_{\alpha}(\Theta_t\lVert\Theta_t')}$ will be. Therefore when the Rényi privacy loss $R_{\alpha}$ is large, the bound for the rate of privacy loss in Eq (10) Lemma 2 will also be smaller (under $(1-\gamma)>0$).
> - **$\gamma$** This is a tuning constant to balance the privacy growth rate estimated using the above two terms, thus helping us tune the privacy loss accumulation. See [Q6]-The gap is constant for more details.
>
> To summarize, the term $\frac{\alpha S_g^2}{4\sigma^2n^2}$ bounds the worst-case privacy loss growth due to noisy gradient update, while the term $\frac{I_{\alpha}(\Theta_t\lVert\Theta_t')}{E_{\alpha}(\Theta_t\lVert\Theta_t')}$ amplifies our bound for the rate of privacy loss, as the Rényi privacy loss accumulates during the process.
>
>
>
> **[Q5b]** The Rényi divergence $R_\alpha$ grows monotonically with its order $\alpha$, as proved in proposition 3 of Mironov [17]. Therefore $\frac{\partial R_\alpha}{\partial \alpha}\geq 0$. Taking this into Lemma 3, we see that the term $I_\alpha/E_\alpha$ is strictly lower bounded by $R_{\alpha}$. Therefore we say $I_{\alpha}/E_{\alpha}$ monotonically increase with $R_\alpha$. The monotonicity here means: under fixed $\alpha$, $\Theta_t$ and $\Theta_t'$, the larger $R_{\alpha}$ is, the larger the lower bound of $I_{\alpha}/E_{\alpha}$ is. We will add a detailed clarification for this monotonicity in Line 172.
>
>
> **[Q6]** Yes, you are right about the gap between our upper bound and lower bound.
>   - **The gap in exponent:** There is a $\frac{2-\eta}{2}$ multiplicative gap between the exponent of our privacy upper bound and the lower bound. In hindsight, this is because discretized noisy GD converges to a biased stationary distribution. Therefore, our LSI constant bound $c=\frac{2-\eta}{2\sigma^2}$ depends on the discretization bias caused by step-size $\eta$, thus causing the exponent gap in our privacy bound.
>   - **The gap in constant:** Our upper bound is larger than the lower bound by roughly a multiplicative constant of two. This is due to the **balancing ratio $\gamma>0$** in Eq (10) of Lemma 2 for bounding the rate of privacy loss.
>     - **At the start of Noisy GD**, setting $\gamma=1$ in Eq (10) results in a smaller privacy loss rate bound. This is because, at the start of noisy GD, the accumulated privacy loss $R_{\alpha}(\Theta_t\lVert\Theta_t')$ is small, thus leading to a small second term $I_{\alpha}/E_{\alpha}$ in Eq (10), by Lemma 3. Setting $\gamma=1$ reduces the coefficient $\frac{1}{\gamma}$ for the dominating first term of Eq(10), at a small cost of increasing the coefficient for the smaller second term $I_{\alpha}/E_{\alpha}$. This facilitates a smaller privacy loss rate bound, and is reflected in the similar growth of composition bound (equivalent to setting $\gamma=1$) and our lower bound in Figure 2.
>     - **As Noisy GD converges**, setting $\gamma\rightarrow 0$ in Eq(10) results in a smaller privacy loss rate bound. This is because, at convergence, the accumulated privacy loss $R_{\alpha}(\Theta_t\lVert\Theta_t')$ is larger, thus leading to more significant second term $I/E$ in Eq (10). Setting $\gamma\rightarrow 0$ in Eq(10) reduces the coefficient $-(1-\gamma)$ for the dominating second term $I/E$, thus facilitate a smaller bound for the privacy loss rate.
>     - In our proof for Theorem 2, we set $\gamma=\frac{1}{2}$ to **balance** privacy loss rate estimates at the start and convergence of noisy GD, thus obtaining the smallest privacy bound at convergence, as shown in Line 610.
>   - **Baseline composition analysis:** Abadi et al.[1] introduce the moments' accountant $\alpha(\lambda)$ for noisy SGD in Eq (2), which effectively tracks the scaled Renyi divergence between processes. Therefore in Figure 1, we plot moment accountant bound in Abadi et al.[1] as baseline composition privacy analysis.
>     - We first use **moments bound on the Gaussian mechanism** (following Lemma 3 in Abadi et al.[1] ) to bound the log moment $\alpha_{\mathcal{M}}(\lambda)$ of data-sensitive computation one update $M: \mathcal{M}(D) = \frac{\eta}{n} \sum\_{x_i\in D} \nabla \ell(\theta;x_i) + \mathcal{N}( 0 , 2\eta\sigma^2 \mathbb{I}\_d )$ in our Algorithm 1.
>       - By  Eq (2) Abadi et al. [1], and that $M(D),M(D')$ are Gaussian distributions (with variance $2\eta\sigma^2$ in every dimension and means at most $\frac{\eta}{n} S_g$ apart in $\ell_2$ norm), we bound $\alpha_{\mathcal{M}}(\lambda) \leq \frac{\lambda (\lambda+1) \eta S_g^2}{4n^2\sigma^2}$.
>     - We then **compose log moment bound for $K$ iterations** by Theorem 2 [Composibility] of log moement bound in Abadi et al.[1]., and we obtain $\alpha(\lambda)\leq K \cdot \alpha_{\mathcal{M}}(\lambda)=\frac{K\lambda(\lambda+1)\eta S_g^2}{4n^2\sigma^2}$.
>     - Finally by definition of log moment (Eq (2) Abadi et al.[1]) and Renyi divergence (Eq (1) in our paper), we take $\lambda \leftarrow \alpha-1$ and $R_{\alpha}(\Theta_K\lVert\Theta_K) \leftarrow \frac{\alpha(\lambda)}{\lambda}$, and obtain the **baseline composition privacy bound** $\epsilon = \frac{\alpha S_g^2}{4n^2\sigma^2}\cdot \eta K$ from the log moment bound.

---

> ### Author Response · Authors · 2021-08-10
> **Response 1 to reviewer wSUc (Q1, Q2, Q3, Q4)**
>
> **[Q1]** Our privacy erosion dynamics analysis for noisy GD is motivated by the rapid convergence analysis for ULA in Vempala and Wibisono [21]. However, because our algorithm noisy GD is slightly different from ULA and we focus on the privacy context, there are many **technical differences** between our approach and that of [21].
> - **Different divergence objective analyzed:** In [21], they analyze **convergence guarantee** of ULA process, and bound the Rényi divergence between the **$k$-th step distribution in ULA** and its **biased stationary distribution**. However, our paper analyzes the **privacy bound**, and bounds a **different** Rényi divergence between the **$k$th step distributions** of **two noisy GD processes** running on neighboring datasets $D,D'$.
>   - The Rényi divergence analyzed in [21] is between one ULA process and its biased stationary distribution, which is **different from** the Rényi privacy loss Eq (1) analyzed in our paper. As a result, we prove converging privacy loss **without assuming or utilizing the convergence of individual noisy GD process** to its stationary distribution in Rényi divergence, while this assumption is made and proved in [21].
>   - The Rényi divergence analyzed in [21] only involves **one evolving ULA process** and a **static stationary distribution** of ULA. However, the privacy loss divergence analyzed in our paper analyzes **two evolving noisy GD processes** running on neighboring datasets $D,D'$. This is technically more difficult than [21] because we use the SDEs for tracing diffusions of two simultaneously evolving noisy GD processes.
>   - The convergence of privacy loss divergence analyzed in our paper requires **a slightly different LSI constant condition** than that in [21]. The convergence guarantee of one ULA process in [21, Lemma 8] only requires LSI constant bound for the **biased stationary distribution**. On the contrary, our privacy analysis requires LSI constant bound **throughout the tracing diffusion processes**, for two noisy GD processes running on neighboring datasets $D,D'$. Our LSI constant condition throughout tracing diffusion is different, as well as more difficult to satisfy, than the LSI condition for stationary distribution in [21].
> - **Different algorithms:** We analyze **Projected Noisy GD algorithm**, while [21] analyzes **ULA** (equivalent to Noisy GD **without projection**). Due to the additional projection operation, our tracing diffusion is semi-continuous, while the underlying diffusion in [21] for ULA updates is continuous. Therefore, our divergence analysis handles the additional complexity due to the **semi-continuity in the tracing diffusion**.
>
>
> To conclude, we summarize our **novelties** in comparison with [21] as follows.
> - **Novel privacy bound for projected noisy GD in Rényi divergence**
>   - We prove **a novel bound for the rate of privacy loss** in Lemma 2 for the tracing diffusion processes underlying projected noisy GD algorithms, which is not proved in [21]. Please see the answer to [Q4] for more details regarding novelty.
>   - We prove **a novel converging privacy bound for noisy GD** in Theorem 2 under LSI condition and smoothness of loss function, **without assuming that the noisy GD process converges to a stationary distribution**. (We later found that **smoothness is not strictly required**, as explained in our answer to Q8.) However, the convergence guarantee of [Lemma 8, 21] requires smoothness of the loss function, the LSI assumption, and that the ULA process converge to a stationary distribution.
>   - We **additionally deal with projection operations** in noisy GD algorithm, and prove **novel privacy bounds and utility bounds** (under smooth strongly convex loss functions). However, the ULA algorithm analyzed in [21] does not contain any projection operations.
> - In Lemma 5, We prove a **novel LSI constant bound** for distributions **throughout** the tracing diffusion of projected noisy GD, under smooth strongly convex loss functions. Meanwhile, [21] assumes LSI constant bound only for the **stationary distribution** of ULA, and does not prove this LSI condition under any practical settings.
>
>
> **[Q2]** Yes, we missed a minus sign.
>
> **[Q3]** Under the condition $\Theta_{\eta k} = \theta_k$, the SDE in Eq (5)
> $$
>   d\Theta_t = - \nabla \mathcal L_D(\theta_k)dt + \sqrt{2\sigma^2}dW_t
> $$
>   is an Ito martingale process
> $$
>   dX_t = \mu(X_t, t)dt + \sigma(X_t, t)dW_t,
> $$
>   where the drift vector $\mu(X_t, t) = - \nabla \mathcal L_D(\theta_k)$ is a $d$-dimentional constant vector, and the diffusion coefficient matrix $D(X_t, t) = \sigma^2 \mathbb I$ is constant matrix. According to Fokker-Planck equation for higher dimensions, the probability density $ p(\theta, t)$ along an Ito process satisfies:
> $$
> \frac{\partial p(x,t)}{\partial t} = -\nabla \cdot \left[\mu(x, t) p(x, t)\right] + \Delta \left[ D(x, t)p(x,t) \right],
> $$
>   where $\nabla \cdot (\cdot）$ is the divergence operator and $\Delta(\cdot)$ is the Laplacian operator defined in Eq (38) and (37) respectively. Since SDE Eq (5) corresponds to the conditional random variable $\Theta_t | \Theta_{\eta k} = \theta_k$, this Fokker Planck equation gives us the equation described in line 127:
> $$
>   \frac{\partial p_{t|\eta k}(\theta| \theta_k)}{\partial t} = \nabla \cdot\left(p_{t|\eta k}(\theta|\theta_k) \nabla \mathcal L_D(\theta_k)\right) + \sigma^2 \Delta p_{t|\eta k}(\theta|\theta_k).
> $$
>   The missing minus sign in (9) is a typo and we’ll fix it.
>
>
> **[Q4]** Yes, our derivation of the rate of Rényi privacy loss for Langevin diffusion is new and novel.
>   - Other than our work, we are only aware of a similar entropy production formula ( Lemma 3.1 of Olla et al.[c], originated from Yau [d]) for KL divergence, which bounds the rate of KL divergence between any two stochastic processes, with terms determined by the partial derivative of base probability density with regard to time $t$. This shares a similar form with our Eq (10).
>   - Our novelty is to calculate this rate of Rényi divergence, with arbitrary order $\alpha$, for Langevin diffusion processes.

---

### Decision · Program_Chairs · 2021-09-27

**Decision:**

Accept (Spotlight)

**Comment:**

There were a lot of productive discussion after the initial reviews. All reviewers are happy with the results in the paper and find the paper an excellent contribution to the problem of differentially private learning.

In particular, instead of the standard approach which shows privacy loss composes over multiple rounds, this paper provides a highly interesting new analysis that shows the Renyi-DP function (up to some order of \alpha) converges exponentially to some fixed value so that more rounds do not introduce additional privacy loss if we only release the last iterate. While the results apply to only the cases when we make some of the nicest assumptions (e.g., strong convexity + strong smoothness), it is the first result of its kind and is significantly stronger comparing to existing attempts to address this niche problem, i.e., privacy amplification by iterations.

Overall I think it's a good paper with original ideas that many readers will pick up.